# ManiPose: Manifold-Constrained Multi-Hypothesis 3D Human Pose Estimation

**Cédric Rommel**[1]     Victor Letzelter[1,3]     Nermin Samet[1]     Renaud Marlet[1,5]

Matthieu Cord[1,2]     Patrick Pérez[1]     Eduardo Valle[1,4]

[1]Valeo.ai, Paris, France   [2]Sorbonne Université, Paris, France
[3]LTCI, Télécom Paris, Institut Polytechnique de Paris, France
[4]Recod.ai Lab, School of Electrical and Computing Engineering, University of Campinas, Brazil
[5]LIGM, Ecole des Ponts, Univ Gustave Eiffel, CNRS, Marne-la-Vallee, France

## Abstract

We propose *ManiPose*, a manifold-constrained multi-hypothesis model for human-pose 2D-to-3D lifting. We provide theoretical and empirical evidence that, due to the depth ambiguity inherent to monocular 3D human pose estimation, traditional regression models suffer from pose-topology consistency issues, which standard evaluation metrics (MPJPE, P-MPJPE and PCK) fail to assess. ManiPose addresses depth ambiguity by proposing multiple candidate 3D poses for each 2D input, each with its estimated plausibility. Unlike previous multi-hypothesis approaches, ManiPose forgoes generative models, greatly facilitating its training and usage. By constraining the outputs to lie on the human pose manifold, ManiPose guarantees the consistency of all hypothetical poses, in contrast to previous works. We showcase the performance of ManiPose on real-world datasets, where it outperforms state-of-the-art models in pose consistency by a large margin while being very competitive on the MPJPE metric.

## 1 Introduction

We propose *ManiPose*, a novel approach for human-pose 2D-to-3D lifting. ManiPose directly addresses the depth ambiguity inherent to monocular 3D human pose estimation by being both multi-hypothesis and manifold-constrained, thus avoiding pose consistency issues, which plague traditional regression-based methods. Unlike previous multi-hypothesis approaches, ManiPose forgoes the use of costly generative models, while still estimating the plausibility of each hypothesis.

Monocular 3D human pose estimation (HPE) is a challenging learning problem that aims to predict 3D human poses given an image or a video from a single camera. Often, the problem is split into two successive steps: first 2D human pose estimation, then 2D-to-3D lifting. Such separation is favorable because 2D-HPE is much more mature, leading to better overall results. Due to depth ambiguity and occlusions, 2D-to-3D lifting is intrinsically ill-posed: multiple 3D poses correspond to the same projection observed in 2D. Despite that, the field has experienced fast developments, with substantial improvements in terms of mean-per-joint-prediction error (MPJPE) and derived metrics (*e.g.*, P-MPJPE, PCK) [52, 53, 42, 47].

However, recent studies [49, 12, 40] noted that poses predicted by state-of-the-art models fail to respect basic invariances of human morphology, such as bilateral sagittal symmetry, or constant length across time of rigid body segments connecting the joints. Not only do we address those concerns with ManiPose (see Fig. 1), but we also provide theoretical elements clarifying the cause of those issues. We show in particular that pose consistency and traditional performance metrics (such as MPJPE)

38th Conference on Neural Information Processing Systems (NeurIPS 2024).

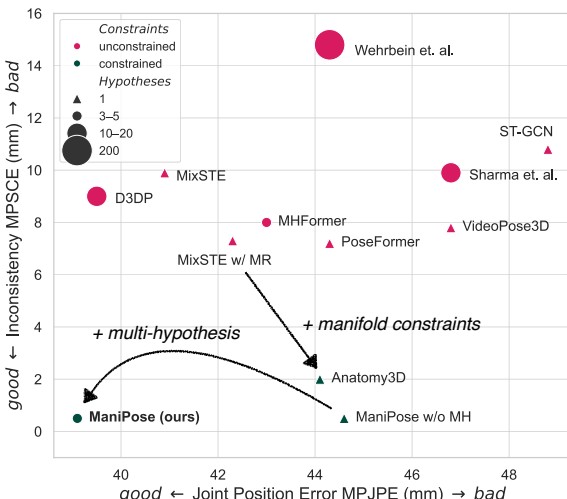

Figure 1: **Optimizing both 3D position and pose consistency requires combining constraints and multiple hypotheses.** Results from Tables 2 and 4. Previous unconstrained methods provide inconsistent poses (top). Regularization (MR) and disentanglement constraints improve consistency, but degrade joint position error (bottom-right). Ours is the only method that achieves both good joint error and consistency, thanks to a combination of disentanglement and a few hypotheses (see circles sizes).

cannot be optimized simultaneously by a standard regression model, because MPJPE ignores the topology of the space of human poses, and traditional regression models imply unimodality, thus overlooking the inherently ambiguous nature of 3D-HPE.

Our contributions include:

- ManiPose, a novel, multi-hypothesis, manifold-constrained model for human-pose 2D-to-3D lifting, which is able to estimate the plausibility of each hypothesis without resorting to costly generative models.

- Theoretical insights that elucidate why traditional regression models associated with standard metrics such as MPJPE fail to enforce pose consistency.

- Extensive empirical results, including comparison to strong baselines, evaluation on two challenging datasets (Human 3.6M and MPI-INF-3DHP), and ablations. ManiPose outperforms state-of-the-art methods by a substantial margin in terms of pose consistency, while still beating them in the MPJPE metric. The ablations confirm the importance of both multiple hypotheses and of constraining the poses to their manifold.

The PyTorch [37] implementation of ManiPose and code used for all our experiments can be found at https://github.com/cedricrommel/manipose.

## 2   Related work

**Regression-based 2D-to-3D pose lifting.** While 2D-to-3D human pose lifting was initially restricted to static frames [31, 3], the field embraced recurrent [13], convolutional [38] and graph neural networks [2, 55, 14, 51] to handle motion. Spatial-temporal transformers appear more recently [42, 53], including MixSTE [52], arguably becoming the state of the art. We adopt them in our work. A few previous works constrain predicted poses to respect human symmetries [50, 4], an idea we advance with a novel constraint implementation, in a multi-hypothesis setting.

**SMPL-based methods.** While 3D human pose lifting's objective is to predict 3D joint positions based on 2D keypoints, the neighboring field of human pose and shape reconstruction (HPSR) aims at estimating whole 3D body meshes from images. HPSR is hence more challenging than 3D-HPE, which explains why models are often larger, frame-based and more reliant on optimization-based post-processing [16, 39, 46, 9]. Nonetheless, our work shares some ideas from this field. Indeed, modern HPSR methods often predict joint angles (and body shape parameters), which are fed to the pre-trained parametric model SMPL [29] to produce human body meshes, thus ensuring that limbs' sizes remain constant along a movement. Note, however, that these are also single-hypothesis regression methods and hence share the same caveats as most 3D-HPE approaches.

**Multi-hypothesis 3D-HPE.** The intrinsic depth-ambiguity of 3D-HPE led the community to investigate multi-hypothesis approaches, including Mixture Density Networks [25, 36, 1], variational autoencoders [44], normalizing flows [18, 49] and diffusion models [12, 6, 10]. Contrary to ours, those methods rely on a generative model to sample 3D pose hypotheses conditioned on the 2D input. A notable exception is MHFormer [27], which, like ManiPose, is deterministic, but treats the hypotheses as intermediate representations to be aggregated at the final network layers, thus concluding with a one-to-one 2D-to-3D mapping. We strive to avoid such injectivity and to preserve the multiple hypotheses, for reasons we will justify both empirically and theoretically in the next sessions. Moreover, none of the previous multi-hypothesis approaches constrain hypotheses to lie on the human pose manifold, thus failing to guarantee good pose consistency.

**Multiple choice learning (MCL)** [11] is a simple approach for estimating multimodal distributions, suited for ambiguous tasks, using the winner-takes-all loss. Adapted for deep learning by Lee *et al.* [20, 21], it produces diverse predictors, each specialized in a particular subset of the data distribution. MCL has proved its effectiveness in several computer vision tasks [41, 19, 33, 8, 30, 45], and was first applied to 2D-HPE in [41]. Our work is the first to employ MCL for the 3D-HPE task, by leveraging recent innovations of Letzelter *et al.* [22].

## 3 ManiPose

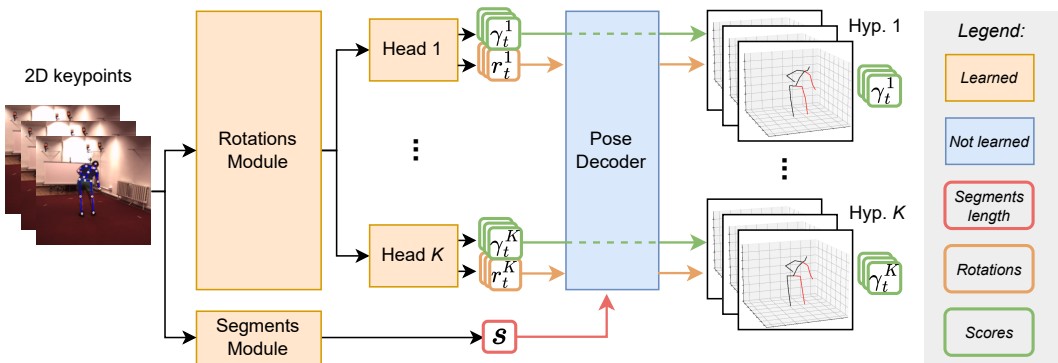

Figure 2: **Overview of ManiPose.** The rotations module predicts $K$ possible sequences of segment rotations with their corresponding likelihoods (scores), while the segments module estimates the shared segment lengths. Hence, predicted poses are constrained to a manifold defined by the estimated lengths, guaranteeing their consistency.

Following the previous state of the art, we split 3D-HPE into two steps, first estimating $J$ human 2D keypoints in the pixel space from a sequence of $T$ video frames $[x_1, \dots, x_T] \in \mathbb{R}^{2 \times J \times T}$, and then lifting them to 3D joint positions $[\hat{p}_1, \dots, \hat{p}_T] \in \mathbb{R}^{3 \times J \times T}$. We focus on the second step (*i.e.*, lifting) in the rest of the paper, assuming the availability of 2D keypoints $x_i$. Our method aims to both ensure pose consistency and resolve depth ambiguity, as we will discuss in the next section.

### 3.1 Constraining predictions to the pose manifold

**Rationale.** Human morphology prevents the joints from arbitrarily occupying the whole space. Instead, the poses within a movement are restricted to a manifold, reflecting the human skeleton's rigidity. If we knew the length of each segment connecting pairs of joints for a given subject, we could guarantee that the predicted poses lie on the correct pose manifold by only predicting the body part's rotations with respect to a reference skeleton. Since we do not have access to ground-truth segment lengths in real use cases, we propose to predict them, thus disentangling the estimation of the reference lengths (fixed across time) from the estimation of the joint rotations (variable across time).

**Disentangled representations.** We constrain model predictions to lie on an estimated manifold by predicting parametrized disentangled transformations of a reference pose $u \in (\mathbb{R}^3)^J$, for which all segments have unit length. Namely, we propose to split the network into two parts (*cf.* Fig. 2):

1. **Segments module**, which predicts segment lengths $s \in \mathbb{R}^{J-1}$, shared by the $T$ frames (time steps) of the input sequence;

2. **Rotations module**, which predicts the rotation $r = [r_{1,0}, \ldots, r_{T,J-1}] \in (\mathbb{R}^d)^{J \times T}$ of each joint relative to their parent joint at each time step.

**Rotations representation.** We represent rotations using 6D continuous embeddings (*i.e.*, $d = 6$). Compared to quaternions or axis-angles, those representations are continuous and, hence, better learned by neural networks, as demonstrated by their proposers [54].

**Pose decoding.** To deliver pose predictions in $(\mathbb{R}^3)^{J \times T}$, the intermediate representations $(s, r)$ must be decoded. We achieve that in three steps (*cf.* Fig. 3):

1. We scale the unit segments of the reference pose $u \in (\mathbb{R}^3)^J$ using $s$, forming a scaled reference pose $u'$: $u'_j = u'_{\tau(j)} + s_j(u_j - u_{\tau(j)})$ for $0 < j \leq J - 1$, where $\tau$ maps the index of a joint to its parent's, if any.

2. For each time step $1 \leq t \leq T$ and joint $0 \leq j < J$, we convert the predicted rotation representations $r_{t,j}$ into rotation matrices $R_{t,j} \in \mathrm{SO}(3)$ (Algorithm 1).

3. We apply those rotation matrices $R_{t,j}$ at each time step $t$ to the scaled reference pose $u'$ using forward kinematics (Algorithm 2).

## 3.2 Multiple choice learning

**ManiPose architecture.** As explained in the introduction, the inherent depth ambiguity of pose lifting requires multiple hypotheses to conciliate pose consistency and MPJPE performance. To address this, we adopt the multiple choice learning (MCL) [21] framework, more precisely leveraging the *resilient MCL* approach as proposed by Letzelter *et al.* [22]. This methodology allows the estimation of conditional distributions for regression tasks, enabling our model to predict multiple plausible 3D poses for each 2D input. Specifically, instead of a single rotation $r_t \in (\mathbb{R}^d)^J$ per time step, ManiPose's rotations module predicts an intermediate representation $e_t \in (\mathbb{R}^{d'})^J$ that feeds $K$ linear heads (with weights $W_r^k$ and $W_\gamma^k$), each predicting its own rotation hypothesis $r_t^k \in (\mathbb{R}^d)^J$ with a corresponding likelihood $\gamma_t^k \in [0, 1]$. That is, for all $1 \leq t \leq T$, $r_t^k = W_r^k e_t$ and $\gamma_t^k = \sigma[\tilde{\gamma}_t]_k$, where the softmax function $\sigma$ is applied to the vector $\tilde{\gamma}_t = [\tilde{\gamma}_t^1, \ldots, \tilde{\gamma}_t^K] \in \mathbb{R}^K$ of intermediate values $\tilde{\gamma}_t^k = W_\gamma^k e_t$.

All rotation hypotheses are decoded together with the shared segment-length predictions $s$, resulting in $K$ hypothetical pose sequences $\hat{p}^k = (\hat{p}_t^k)_{t=1}^T$, with corresponding likelihood sequences $\gamma^k = (\gamma_t^k)_{t=1}^T$, called **scores** hereafter (Fig. 2).

**Loss function.** As in [22], ManiPose is trained with a composite loss

$$\mathcal{L} = \mathcal{L}_{\mathrm{wta}} + \beta \mathcal{L}_{\mathrm{score}} . \tag{1}$$

The first term, $\mathcal{L}_{\mathrm{wta}}$, is the winner-takes-all loss [21]

$$\mathcal{L}_{\mathrm{wta}}(\hat{p}(x), p) = \frac{1}{T} \sum_{t=1}^T \min_{k \in [\![1,K]\!]} \ell(\hat{p}_t^k(x), p_t) , \tag{2}$$

where $\ell(\hat{p}_t^k(x), p_t) \triangleq \frac{1}{J} \sum_{j=0}^{J-1} \|p_{t,j} - \hat{p}_{t,j}^k(x)\|_2$, and $\hat{p}_t^k(x)$ denotes the pose prediction at time $t$ using the $k^{\mathrm{th}}$ head. The second term, $\mathcal{L}_{\mathrm{score}}$, is the scoring loss

$$\mathcal{L}_{\mathrm{score}}(\hat{p}(x), \gamma(x), p) = \frac{1}{T} \sum_{t=1}^T \mathcal{H}\big(\delta(\hat{p}_t, p_t), \gamma_t(x)\big) , \tag{3}$$

where $\mathcal{H}(\cdot, \cdot)$ is the cross-entropy, $\hat{p}_t = (\hat{p}_t^k)_{k=1}^K$, and

$$[\delta(\hat{p}_t, p_t)]_k \triangleq \mathbf{1}\Big[k \in \operatorname*{arg\,min}_{k' \in [\![1,K]\!]} \ell\left(\hat{p}_t^{k'}, p_t\right)\Big] \tag{4}$$

is the indicator function of the *winner* pose hypothesis, which is the closest to the ground truth. Eq. (3) is the average cross-entropy between target and predicted scores $\gamma_t(x) \in [0,1]^K$ at each time $t$.

Those losses are complementary. The winner-takes-all loss updates only the best predicted hypothesis, specializing each head on part of the data distribution [21]. The scoring loss allows the model to learn how likely each head is to winning, thus avoiding overconfidence of non-winner heads (*cf.* [19, 45]).

**Conditional distribution estimation.** As detailed in [22], the model may be interpreted probabilistically as a multimodal conditional density estimator. More precisely, it models the distribution $P(p|x)$ of 3D poses conditioned on 2D poses as a mixture of Dirac distributions:

$$\hat{P}(p|x) \triangleq \sum_{k=1}^{K} \gamma^k(x) \delta_{\hat{p}^k(x)}(p) \,. \tag{5}$$

Hence, the predicted conditional distribution has, at each predicted hypothesis $\hat{p}^k$, a peak whose likelihood is given by the predicted score $\gamma^k$. As described in Section 4, interpreting hypotheses and scores probabilistically enables us to handle depth ambiguity.

# 4 Formal analysis

ManiPose, as outlined in Section 3, is crafted to address the flaws inherent in unconstrained, single-hypothesis lifting-based 3D-HPE methods (see Fig. 1). This section illustrates that without ManiPose's critical components (multiple hypotheses and manifold constraint), it is impossible to simultaneously minimize joint error and ensure pose consistency (Section 4.1). To illustrate this, a toy example within a simplified 1D-to-2D framework is provided in Section 4.2.

Figure 3: **Pose decoder overview.**

## 4.1 Single-hypothesis position-error minimization leads to inconsistent skeleton lengths

We formally highlight the limitations of unconstrained single-hypothesis 3D-HPE, justifying our approach, which combines consistency constraints and multiple hypotheses to resolve depth ambiguity.

Let $p = [p^1, \ldots, p^J] \in \mathbb{R}^{3 \times J}$ be a human pose, defined by the Cartesian 3D coordinates of each of the $J$ joints of a predefined skeleton. Then, a sequence of $T$ poses of the same subject at increasing time steps $t_1 \ldots t_T \in \mathbb{R}$ forms a movement $m = [p_0, \ldots, p_T] \in \mathbb{R}^{3 \times J \times T}$. Assuming bone length is fixed during a movement (which is empirically verifiable in human pose datasets), then the poses $p_t$ of $m$ must all lie on the same smooth manifold.

**Proposition 4.1** (Human pose manifold). *Assuming a rigid skeleton, all poses of a movement* $m = [p_t]_{t=1}^{T}$ *lie on a manifold* $\mathcal{M}$ *of dimension* $2(J-1)$:

$$\forall t \in \{1, \ldots, T\}, \quad p_t \in \mathcal{M} \,. \tag{6}$$

**Proof sketch.** (Detailed in Appendix B). Skeleton rigidity implies that, if $i$ is a joint connected to the root, then it lies on a 2D sphere $S^2(0, s_{i,0})$ centered at the origin with fixed radius $s_{i,0}$. Another joint $j$ linked to $i$ has a position expressible by its spherical coordinates relative to $i$ with fixed radius $s_{j,i}$. That implies an homeomorphism between the position $p_{t,j}$ of joint $j$ and the direct product of spheres centered at the origin $S^2(0, s_{i,0}) \times S^2(0, s_{j,i})$. By induction, one can show that $p_t$ lies on a subspace of $(\mathbb{R}^3)^J$, which is homeomorphic to a product of spheres centered at the origin. ∎

Proposition 4.1 implies that all poses predicted for a video sequence should ideally lie on the same manifold $\mathcal{M}$ as the ground-truth data, which is homeomorphic to the direct product of 2D unit spheres $(S^2)^{J-1}$ (*cf.* Appendix B). Crucially, we can further show that minimizing joint position error using a single-hypothesis model necessarily leads to predicted poses lying outside the true manifold:

**Proposition 4.2** (Inconsistency of MSE minimizer). *With a rigid skeleton and mild assumptions on the training distribution, predicted 3D poses minimizing the traditional mean squared error (MSE) loss lie outside the pose manifold* $\mathcal{M}$.

**Proof sketch.** (See Appendix B). Consider a skeleton with $J$ joints, with $(x, p)$, as pairs of corresponding 2D inputs and 3D poses. Let the function $\ell = (\ell_j)_{j=1}^{J-1}$ compute the lengths of the segments in a pose, which shall remain constant. On a dataset $\{(x_i, p_i)\}_{i=1}^N$ drawn from the joint distribution of 2D and 3D poses, let the expected MSE of a traditional predictive model $f$ be $\mathbb{E}_{x,p}\left[\|p - f(x)\|_2^2\right]$. Let the ideal model $f^*$ be the one minimizing that expected MSE, which is the conditional expectation $f^*(x) = \mathbb{E}[p \mid x]$. Jensen inequality and the rigidity assumption imply that, for any joint $j$, $\ell_j^2\left(f^*(x)\right) < s_j^2$ where $s_j$ is the true length of the segment associated with joint $j$. This shows that the poses predicted by $f^\star$ violate the original segment length constraints, and thus, the original rigidity assumption. ∎

Proposition 4.2 has the following implications:

1. Traditional unconstrained single-hypothesis approaches are bound to predict inconsistent movements, where bone lengths may vary.
2. With a single hypothesis, models constrained to the manifold will always lose to unconstrained models in terms of MPJPE performance (formalized in Corollary B.1).
3. The only way of reaching both optimal MPJPE and consistency is through multiple hypotheses (formalized in Corollary B.3).

Therefore, the MPJPE metric (and its traditional extensions) is insufficient to assess 3D-HPE, as it completely ignores pose consistency. Furthermore, we are able to prove in Appendix B.2 that multiple hypotheses (constrained or not) can always reach better joint position errors than single-hypothesis models.

## 4.2 Insights to the formal argument on a simplified setting

We illustrate the argument of Section 4.1 with a simplified 1D-to-2D setup. We further generalize this intuitive illustration to the 2D-to-3D setting in Appendix C of the supplementary.

As in human pose lifting, we take a root joint $J_0$ as reference, fixed at $(0,0)$. For a joint $J_1$, the problem amounts to predicting the 2D position $(x, y)$, given its 1D projection $u = x$, assuming a constant distance $s = 1$ between them. This simplification ignores the camera perspective and considers the joints to be connected by a rigid segment as in the case of human poses.

Table 1: **1D-to-2D performance.** Fig. 4-D setting, results averaged over five random seeds.

|  | MPJPE ↓ | Distance to circle ↓ |
|---|---|---|
| Unconst. MLP | $0.753 \pm 0.008$ | $0.42 \pm 0.01$ |
| Constrained MLP | $0.777 \pm 0.027$ | $\mathbf{0.00 \pm 0.00}$ |
| ManiPose | $\mathbf{0.752 \pm 0.012}$ | $\mathbf{0.00 \pm 0.00}$ |

We train three different models with comparable architectures on two datasets $\{(x_i, (x_i, y_i))\}_{i=1}^N$ sampled from the angular distributions represented in blue on Fig. 4. The models correspond to:

1. A 2-layer MLP (✖) trained to minimize the mean squared error between true $(x, y)$ and predicted joint positions $(\hat{x}, \hat{y})$;
2. A constrained MLP of the same size (✖), predicting the angle $\hat{\theta}$ instead of the joint position;
3. ManiPose: our constrained multi-hypothesis model capable of predicting $K = 2$ possible angles $(\hat{\theta}^k)_{k=1}^K$ with their corresponding likelihoods.

Fig. 4 shows that the traditional unconstrained single-hypothesis approach (✖) leads to good results in an easy unimodal scenario (C), but fails when facing a more challenging bimodal distribution (D), leading to predictions outside the circle manifold, as depth ambiguity makes the lifting problem ill-posed. The single-hypothesis constrained model (✖) delivers predictions on the circle, at the cost of worse MPJPE performance than the unconstrained MLP. Such performance decrease is due to the Euclidean topology of the MPJPE metric having its minimum (●) outside the manifold (Fig. 4-B).

Crucially, this implies that the unconstrained single-hypothesis models are bound to make inconsistent predictions, with varying "bone lengths" (the circle radius). It also shows that models constrained to the manifold (circle) will always be outcompeted by unconstrained models on MPJPE performance.

Predicting multiple hypotheses constrained to the circle, with their respective likelihoods (★ in Fig. 4-B) allows escaping this dilemma, which is exactly what ManiPose does (▲ in Fig. 4-D). The

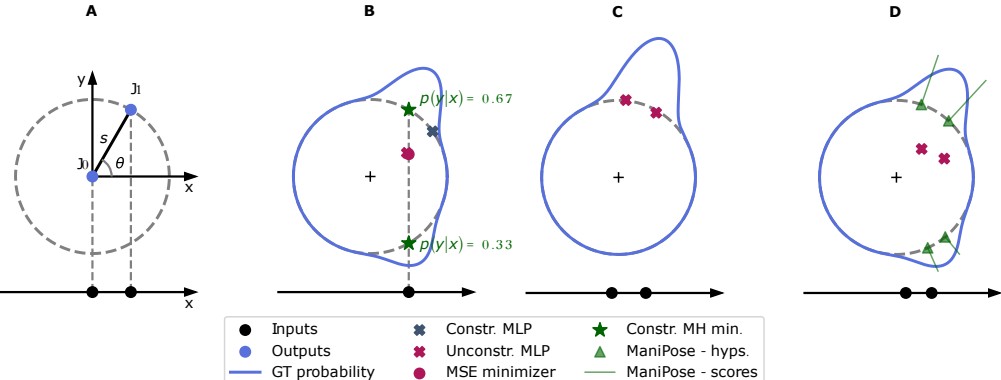

Figure 4: **(A)** 1D-to-2D articulated pose lifting problem. **(B)** True MSE minimizers under a multi-modal distribution. One-to-one mappings cannot both reach optimal performance and stay on the pose manifold (dashed circle). **(C)** Without depth ambiguity, unconstrained models are effective. **(D)** Ambiguity from multimodal distributions challenges both constrained and unconstrained models. Multi-hypothesis approaches can deliver an acceptable solution to the problem.

predicted hypotheses are all on the circle, contrary to the unconstrained MLP, and spread between the two distribution modes, unlike the constrained single-hypothesis method.

Moreover, the predicted scores (length of green lines) match the $\frac{2}{3}$ and $\frac{1}{3}$ ground-truth likelihoods of the two modes. Those advantages translate into perfect pose consistency and into comparable MPJPE performance with respect to the unconstrained MLP (Table 1).

## 5 Experiments

### 5.1 Experimental setup

**Datasets.** We evaluate our model on two 3D-HPE datasets. **Human 3.6M** [15] contains 3.6 million images of 7 actors performing 15 different indoor actions. It is the most widely used dataset for 3D-HPE. Following previous works [52, 27, 53, 38], we train on subjects S1, S5, S6, S7, S8, and test on subjects S9 and S11, adopting a 17-joint skeleton (*cf.* Fig. 5). We employ a pre-trained CPN [5] to compute the input 2D keypoints, as in [38, 52]. **MPI-INF-3DHP** [32] also adopts a 17-joint skeleton, but, with fewer samples and containing both indoor and outdoor scenes, it is more challenging than Human 3.6M. We used ground-truth 2D keypoints for this dataset, as usually done [53, 4, 52].

**Traditional evaluation metrics.** The mean per-joint position error (MPJPE) is the usual performance metric for the datasets above, under different protocols, both reported in mm. In protocol #1, the root joint position is set as a reference, and the predicted root position is translated to 0. In protocol #2 (P-MPJPE), predictions are additionally Procrustes-corrected. For MPI-INF-3DHP, additional thresholded metrics derived from MPJPE are often reported, such as AUC (Area Under Curve) and PCK (Percentage of Correct Keypoints) with a threshold at 150 mm, as explained in [32].

**Pose consistency metrics.** MPJPE being insufficient to assess pose consistency (Section 4), we further assess to which extent predicted skeletons are rigid by measuring the average standard deviations of segment lengths across time in predicted action sequences:

$$\text{MPSCE} \triangleq \frac{1}{J-1} \sum_{j=1}^{J-1} \sqrt{\frac{1}{T} \sum_{t=1}^{T} (s_{t,j,\tau(j)} - \bar{s}_{j,\tau(j)})^2}, \tag{7}$$

with $s_{t,j,i} = \|\hat{\mathrm{p}}_{t,j} - \hat{\mathrm{p}}_{t,i}\|_2$ and $\bar{s}_{j,i} = \frac{1}{T} \sum_{t=1}^{T} s_{t,j,i}$, where $\tau$ was defined in Section 3.1. We call this metric, reported in mm, the Mean Per Segment Consistency Error (MPSCE).

Following [12, 40], we also assess the bilateral symmetry of predicted skeletons through the Mean Per Segment Symmetry Error (MPSSE), in mm:

$$\text{MPSSE} \triangleq \frac{1}{T \, |\mathcal{J}_{\text{left}}|} \sum_{t=1}^{T} \sum_{j \in \mathcal{J}_{\text{left}}} \left| s_{t,j,\tau(j)} - s_{t,j',\tau(j')} \right|, \qquad \text{with} \quad j' = \zeta(j), \tag{8}$$

where $\mathcal{J}_{\text{left}}$ denotes the set of indices of left-side joints and $\zeta$ maps left-side joint indices to their right-side counterparts.

**Multi-hypothesis evaluation protocol.** One must decide how to use multiple hypotheses to compute the metrics. The dominant approach [24, 25, 36, 44, 49, 12] is the **oracle** evaluation, *i.e.*, using the predicted hypothesis closer to the ground truth (*i.e.*, Eq. (2) for MPJPE). That makes sense for multi-hypothesis methods, as the oracle metric measures the distance between the target and the discrete set of predicted hypotheses. It aligns with the idea of many possible outputs for a given input.

Hypotheses can also be *aggregated* into a final pose, *e.g.*, through unweighted or weighted averaging (using predicted scores). The latter has the disadvantage of falling back to a one-to-one mapping scheme, which is precisely what we want to avoid in a multi-hypothesis setting.

We report both oracle and aggregated metrics in our experiments, favoring oracle results.

**Implementation details.** ManiPose, as presented in Section 3, is compatible with any backbone. Here, we chose to build on the MixSTE [52] network for both the rotations and the segment modules (the latter in a reduced scale). Details about our architecture and training appear in Appendix D.

## 5.2  Comparison with the state of the art

Table 2: **Pose consistency evaluation of state-of-the-art methods on Human3.6M.** MPJPE performance and pose consistency are not correlated; only ManiPose excels in both. $T$: sequence length. $K$: number of hypotheses. Orac.: Metric computed using oracle hypothesis. Grey lines: Methods where the Oracle MPJPE is computed with non-comparable number of hypotheses with respect to the other baselines. **Bold**: best; Underlined: second best. *: Method with unavailable code ; MPSSE values reported in [12]. †: Results with comparable number of hypotheses. ‡: Results computed with official checkpoint and code.

| | $T$ | $K$ | Orac. | MPJPE↓ | MPSSE↓ | MPSCE↓ |
|---|---|---|---|---|---|---|
| *Single-hypothesis methods:* | | | | | | |
| ST-GCN [2] | 7 | 1 | | 48.8 | 8.9 | 10.8 |
| VideoPose3D [38] | 243 | 1 | | 46.8 | 6.5 | 7.8 |
| PoseFormer [53] | 81 | 1 | | 44.3 | 4.3 | 7.2 |
| Anatomy3D [4] | 243 | 1 | | 44.1 | 1.4 | 2.0 |
| MixSTE [52] | 243 | 1 | | 40.9 | 8.8 | 9.9 |
| *Multi-hypothesis methods:* | | | | | | |
| Wehrbein *et al.* [49] | 1 | 200 | ✓ | 44.3 | 12.2 | 14.8 |
| DiffPose (Holmquist *et al.*) [12]* | 1 | 200 | ✓ | 43.3 | 14.9 | - |
| GFPose [6] | 1 | 200 | ✓ | 35.6 | 13.1 | 16.5 |
| D3DP (P-Best) [43] | 243 | 20 | ✓ | 39.5 | 6.9 | 9.0 |
| GFPose [6]† | 1 | 10 | ✓ | 45.1 | 13.1 | 16.5 |
| Sharma *et al.* [44] | 1 | 10 | ✓ | 46.8 | 13.0 | 9.9 |
| DiffPose (Gong *et al.*) [10]‡ | 243 | 5 | ✓ | 39.3 | 5.2 | 6.1 |
| MHFormer [27] | 351 | 3 | | 43.0 | 5.7 | 8.0 |
| ManiPose (Ours) | 243 | 5 | | 42.1 | 0.4 | 0.8 |
| ManiPose (Ours) | 243 | 5 | ✓ | **39.1** | **0.3** | **0.5** |

**Human 3.6M.** Comparisons with state-of-the-art single- and multi-hypothesis methods are presented in Table 2 and illustrated in Fig. 1. ManiPose outperforms previous methods in terms of Oracle MPJPE in comparable scenarios, while reaching nearly perfect consistency. Moreover, note that MPJPE and consistency metrics are not positively correlated for single-hypothesis methods. As predicted in Section 4.1, our empirical results show that MPJPE improvements achieved by MixSTE come at the cost of poorer consistency compared to previous models. In contrast, the only single-hypothesis constrained model, Anatomy3D [4], achieves good consistency at the expense of inferior MPJPE. Those results empirically validate the theoretical predictions of Sections 4.1 and B, further

confirming what we have shown, intuitively, in the simplified 1D-to-2D setting (Section 4.2). Note that while ManiPose is deterministic, previous multi-hypothesis methods are generative, except for MHFormer. Table 2 shows that they require up to two orders of magnitude more hypotheses than ManiPose to reach competitive performance (see, *e.g.*, the performance of GFPose). This property is expected. Indeed, optimization based on Winner-Takes-All theoretically leads to an optimal coverage of the modes of the conditional distribution with a fixed number of samples [23], in contrast to generative-based approaches. This is reflected in the oracle metric, which approximates the so-called *quantization* (or Distortion) error, as defined in (27), when the number of data points is large. More detailed MPJPE results per action appear in Tables 8 and 9 in the supplemental. We also complement our analysis on the diversity of ManiPose in Fig. 11 of the appendix.

Fig. 6 showcases qualitative results, where multiple hypotheses help in depth-ambiguous situations.

Table 3: **Comparison with the state-of-the-art on MPI-INF-3DHP using ground-truth 2D poses.** $T$: sequence length.

|  | $T$ | PCK↑ | AUC↑ | MPJPE↓ | MPSSE↓ | MPSCE↓ |
|---|---|---|---|---|---|---|
| VideoPose3D [38] | 81 | 85.5 | 51.5 | 84.8 | 10.4 | 27.5 |
| PoseFormer [53] | 9 | 86.6 | 56.4 | 77.1 | 10.8 | 14.2 |
| MixSTE [52] | 27 | 94.4 | 66.5 | 54.9 | 17.3 | 21.6 |
| P-STMO [42] | 81 | 97.9 | 75.8 | **32.2** | 8.5 | 11.3 |
| ManiPose (Ours) Aggr. | 27 | 98.0 | 75.3 | 37.7 | **0.6** | **1.3** |
| ManiPose (Ours) Orac. | 27 | **98.4** | **77.0** | 34.6 | **0.6** | **1.3** |

**MPI-INF-3DHP.** Similar results were obtained for this dataset (*cf.* Table 3). Not only does ManiPose reach consistency errors close to 0, but also best PCK and AUC performance. As for MPJPE, only [42] achieves slightly better performance, at the cost of large pose consistency errors.

## 5.3 Ablation study

Table 4: **Ablation study: Single hypothesis cannot optimize both MPJPE and consistency.** ManiPose uses the same backbone as MixSTE. MR: with manifold regularization. MC: manifold-constrained. **Bold**: best. Underlined: second best.

|  | MR | MC | $K$ | # Params. | MPJPE↓ | MPSSE↓ | MPSCE↓ |
|---|---|---|---|---|---|---|---|
| ManiPose (Ours) | ✗ | ✓ | 5 | 34.44 M | **39.1** | **0.3** | **0.5** |
| w/o MH | ✗ | ✓ | 1 | 34.42 M | 44.6 | **0.3** | **0.5** |
| w/o MC, w/ MR | ✓ | ✗ | 1 | 33.78 M | 42.3 | 5.7 | 7.3 |
| w/o MR (MixSTE) | ✗ | ✗ | 1 | 33.78 M | 40.9 | 8.8 | 9.9 |

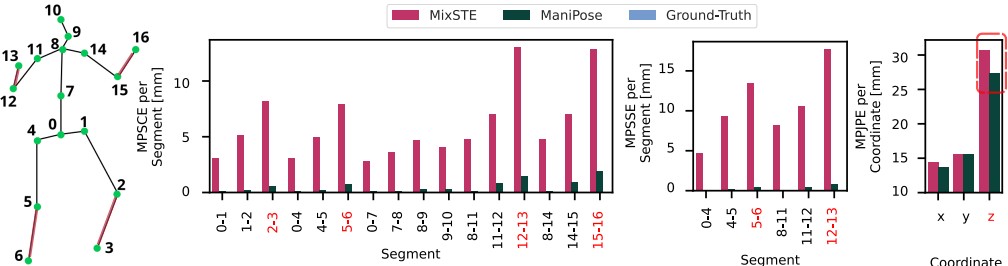

Figure 5: **MPSCE, MPSSE and MPJPE per segment/coordinate (lower is better).** ManiPose mostly helps to deal with the depth ambiguity ($z$ coordinate). Ground-truth poses are represented but not visible because they have perfect consistency.

**Impact of components.** We evaluate the impact of removing each component of ManiPose on the Human3.6M performance (Table 4). The components tested are the multiple hypotheses (MH) and the manifold constraint (MC). We also compare MC to a more standard manifold regularization (MR), *i.e.*, adding Eq. (7) to the loss. Note that without all these components, we fall back to MixSTE [52], and that the performances reported in Table 4 also appear in Fig. 1.

We see that MR helps to improve pose consistency, but not as much as MC. However, without multiple hypotheses, MC consistency improvements come at the cost of degraded MPJPE performance, as

foreseen by our formal analysis (Section 4). Only the combination of both MC and MH allows us to optimize both consistency and MPJPE.

**Fine error analysis.** We can see in Fig. 5 that, compared to MixSTE, ManiPose reaches substantially superior MPSSE and MPSCE, consistency across all skeleton segments. Furthermore, note that larger MixSTE errors occur for segments KNEE-FOOT and ELBOW-WRIST, which are the most prone to depth ambiguity. That agrees with coordinate-wise errors depicted in Fig. 5, showing that ManiPose improvements mostly translate into a reduction of MixSTE depth errors, which are twice as large as for other coordinates. Further ablations, including the effect of the number of hypotheses $K$, the score loss weight $\beta$ and the rotations representation choice appear in the supplemental.

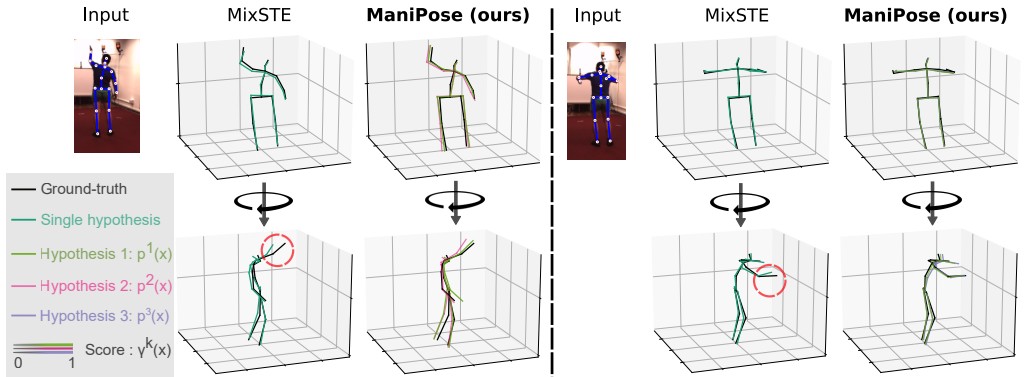

Figure 6: **Qualitative comparison between ManiPose and state-of-the-art regression method, MixSTE.** Two pairs of predicted hypotheses by ManiPose are illustrated in green-pink (left) and green-purple (right), where opacity is used to represent the predicted scores. Multiple hypotheses and constraints help to deal with depth ambiguities and avoids predicting shorter limbs (red circles).

## 6   Conclusion

We presented a new manifold-constrained multi-hypothesis human pose lifting method (ManiPose) and demonstrated its empirical superiority to the existing state-of-the-art on two challenging datasets. Further, we provided theoretical evidence supporting the tenets of our method, by showing the inherent limitation of unconstrained single-hypothesis approaches to 3D-HPE. We established that unconstrained single-hypothesis methods cannot deliver consistent poses and that constraining or regularizing single-hypothesis models leads to worse position errors. We also showed that traditional MPJPE-like metrics are insufficient to assess consistency.

**Limitations.** To guarantee its consistency, ManiPose relies on the forward kinematics algorithm, which is inherently sequential across joints. Removing that dependence is an interesting avenue for accelerating the method. On another note, while ManiPose ensures the rigidity of the predicted poses, imposing constraints within human body articulation limits presents another area for enhancement.

## Acknowledgments and Disclosure of Funding

This work was granted access to the HPC resources of IDRIS under the allocation 2023-AD011014073 made by GENCI. It was also partly funded by the French Association for Technological Research (ANRT CIFRE contract 2022-1854). We are grateful to the reviewers for their insightful comments.

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

# Appendix / supplemental material

This supplemental material is organized as follows:

- Appendix A contains empirical verification of our assumptions,
- Appendix B presents the proofs of our theoretical results, together with a few corollaries,
- Appendix C provides further implementation details concerning the 1D-to-2D experiment, as well as an extension to the 2D-to-3D setting,
- Appendix D contains implementation and training details concerning ManiPose, as well as compared baselines,
- Appendix E presents further results of the Human 3.6M experiment,
- and finally, Appendix F explains the provided experiment code.

## A    Assumption verifications

Let us first define a few elements that we will need needed for our derivations.

**Definition A.1** (Human skeleton). We define a human skeleton as an undirected connected graph $G = (V, E)$ with $J = |V|$ nodes, called *joints*, associated with different human body articulation points. We assume a predefined order of joints and denote $A = [A_{ij}]_{0 \leq i,j < J} \in \{0, 1\}^{J \times J}$ the adjacency matrix of $G$, defining joints connections.

**Definition A.2** (Human pose and movement). Let $G$ be a skeleton of $J$ joints. We attach to each joint $i$ a position $\mathrm{p}_i^G$ in $\mathbb{R}^3$ and call the vector $\mathrm{p}^G = [\mathrm{p}_0^G, \ldots, \mathrm{p}_{J-1}^G] \in (\mathbb{R}^3)^J$ a *human pose*. Furthermore, given a series of increasing time steps $t_1 < t_2 < \cdots < t_T \in \mathbb{R}$, we define a human *movement* m as a sequence of poses *of the same subject* at those instants $\mathrm{m} = [\mathrm{p}_{t_1}^G, \ldots, \mathrm{p}_{t_T}^G] \in (\mathbb{R}^3)^{J \times T}$.

We base the theoretical results of Section 4.1 on the following assumptions. The first states the reference frame traditionally used for assessing 3D-HPE models:

**Assumption A.3** (Reference root joint). For any skeleton $G$ and movement m of length $T$, the joint of index 0, called the *root joint*, is at the origin $\mathrm{p}_{t,0}^G = [0, 0, 0]$ at all times $t_1 \leq t \leq t_T$. That is equivalent to measuring positions $\mathrm{p}_t^G$ in a reference frame attached to the root joint.

The second assumption concerns the rigidity of human body parts:

**Assumption A.4** (Rigid segments). We assume that the Euclidean distance between adjacent joints is constant within a movement m: for any pair of instants $t$ and $t'$ and for any joints $i, j$ such that $A_{ij} = 1$, we assume that

$$s_{t,i,j} = s_{t',i,j} = s_{i,j}, \tag{9}$$

where $s_{t,i,j} = \|\mathrm{p}_{t,i}^G - \mathrm{p}_{t,j}^G\|_2 > 0$.

Finally, we assume that the conditional distribution of poses does not collapse to a single point, *i.e.*, that we have a one-to-many problem:

**Assumption A.5** (Non-degenerate conditional distribution). Given a joint distribution $\mathrm{P}(\mathrm{x}^G, \mathrm{p}^G)$ of 3D poses $\mathrm{p}^G \in (\mathbb{R}^3)^J$ and corresponding 2D inputs $\mathrm{x}^G \in (\mathbb{R}^2)^J$, we assume that the conditional distribution $\mathrm{P}(\mathrm{p}^G | \mathrm{x}^G)$ is non-degenerate, *i.e.*, it is not a single Dirac distribution.

Note that can be true even when $\mathrm{P}(\mathrm{x}^G, \mathrm{p}^G)$ is unimodal (*e.g.*, Fig. 4).

We verified on Human 3.6M [15] ground-truth data that assumptions A.4 and A.5 hold for actual poses in both training and test splits.

**Segments rigidity.** As shown on Figs. 5 and 9, ground-truth 3D poses have perfect MPSSE (8) and MPSCE (7) metrics, meaning that ground-truth skeletons are perfectly symmetric, with rigid segments. Assumption A.4 is thus verified in actual training and test data.

**Non-degenerate distributions.** As shown on Fig. 7, the conditional distribution of ground-truth 3D poses given 2D keypoints position is clearly multimodal, and, thus, non-degenerate (not reduced to a single Dirac distribution). That validates assumption A.5 and explains why multi-hypothesis techniques are necessary.

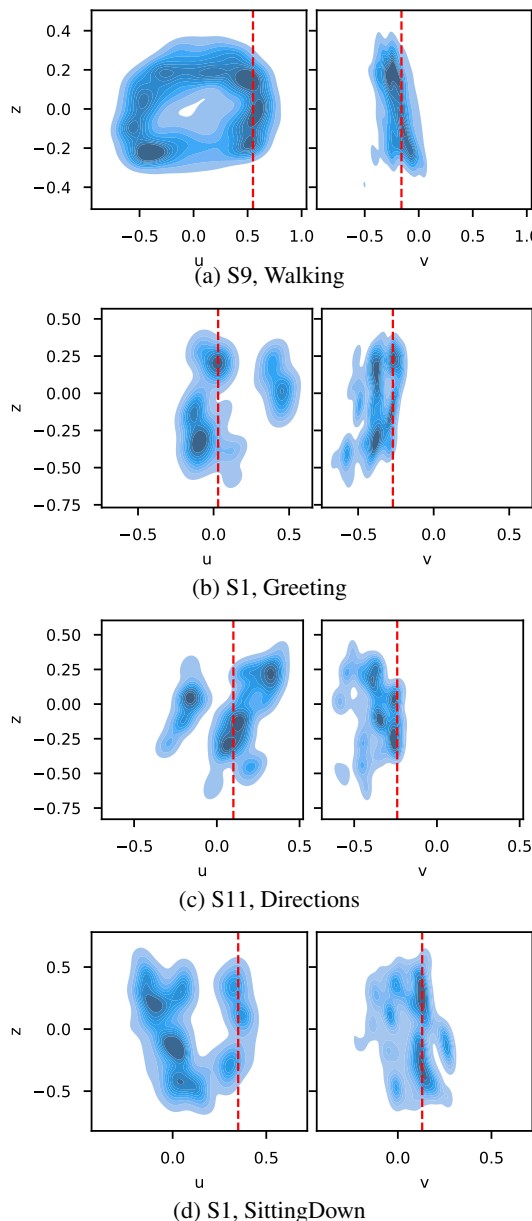

Figure 7: **Estimated joint distributions of ground-truth 2D inputs ($u$, $v$ pixel coordinates) together with 3D $z$-coordinates (depth) for different subjects and actions.** The depth density conditional on inputs is clearly multimodal. Vertical red lines are examples of depth-ambiguous inputs. Distributions are estimated with a kernel density estimator from the Seaborn plotting library [48].

# B  Proofs and additional corollaries

## B.1  Properties of manifold constraint and multi-hypotheses models

This section contains the proofs of the theoretical results presented in Section 4.1, together with a few corollaries.

PROOF. [Proposition 4.1] Let $i$ be a joint connected to the root $p_0$ (*i.e.*, $A_{i0} = 1$). From assumptions A.3 and A.4, we know that at any instant $t$, $\mathrm{p}_{t,i}^G$ lies on the sphere $S^2(0, s_{i,0})$ centered at 0 with radius $s_{i,0}$ independent of time. Therefore, its position can be fully parameterized in spherical coordinates by two angles $(\theta_{t,i}, \phi_{t,i})$. Let $j$ be a joint connected to $i$. Like before, assumption A.4 implies that at any instant $t$, $\mathrm{p}_{t,j}^G$ lies on the moving sphere $S^2(p_{t,i}^G, s_{j,i})$ centered at $p_{t,i}^G$ with radius $s_{j,i}$ independent of time. Thus, we can fully describe $\mathrm{p}_{t,j}^G$ with the position of its center, $p_{t,i}^G$ and the spherical coordinates $(\theta_{t,j}, \phi_{t,j})$ of joint $j$ relative to the center of the sphere, *i.e.*, joint $i$. That means that there is a bijection between the possible positions attainable by $\mathrm{p}_{t,j}^G$ at any instant and the direct product of spheres $S^2(0, s_{i,0}) \otimes S^2(0, s_{j,i})$.[1] That bijection is an homeomorphism since it is a composition of homeomorphisms: we can compute $\mathrm{p}_{t,j}^G$ from $(\theta_{t,i}, \phi_{t,i}, \theta_{t,j}, \phi_{t,j})$ following the forward kinematics algorithm [34] (*cf.* algo. 2), *i.e.*, using a composition of rotations and translations.

Now let us assume for some arbitrary joint $k$ that $\mathrm{p}_{t,k}^G$ lies at all times on a space $\mathcal{M}_{2d}$ homeomorphic to a product of spheres of dimension $2d$. That means that $\mathrm{p}_{t,k}^G$ can be fully parametrized using $2d$ spherical angles $(\theta_1, \phi_1, \ldots, \theta_d, \phi_d)$. Let $l$ be a joint connected to $k$ (typically one further step away from the root joint $p_0$ and not already represented in $\mathcal{M}_{2d}$). As before, at any instant $t$, $\mathrm{p}_{t,l}^G$ needs to lie on the sphere centered on $\mathrm{p}_{t,k}^G$ of constant radius $s_{k,l}$. Thus, we can fully describe $\mathrm{p}_{t,l}^G$ using the $2(d+1)$-tuple of angles obtained by concatenating its spherical coordinates relative to joint $k$, together with the $2d$-tuple describing $\mathrm{p}_{t,k}^G$, *i.e.* the center of the sphere. So $\mathrm{p}_{t,l}^G$ lies on a space $\mathcal{M}_{2(d+1)}$ homeomorphic to a product of spheres of dimension $2(d+1)$.

We can conclude by induction that at any instant $t$, $\mathrm{p}_t = [\mathrm{p}_{t,1}^G, \ldots, \mathrm{p}_{t,J}^G]$ lies on the same subspace of $(\mathbb{R}^3)^J$, which is homeomorphic to a product of spheres centered at the origin:

$$\bigotimes_{i<j/A_{ij}=1} S^2(0, s_{i,j}) \,. \tag{10}$$

Finally, the previous space is trivially homeomorphic to $(S^2)^{J-1}$ through the scaling $(1/s_{i,j})_{i<j/A_{ij}=1}$. $(S^2)^{J-1}$ is a manifold of dimension $2(J-1)$ as the direct product of $J-1$ manifolds of dimension 2. ∎

PROOF. [Proposition 4.2] Let $G$ be a skeleton with $J$ joints, $\mathrm{x} \in (\mathbb{R}^2)^J$ a 2D pose, $\mathrm{p} \in (\mathbb{R}^3)^J$ its corresponding 3D pose, and $\mathrm{P}(\mathrm{x}, \mathrm{p})$ a joint distribution of poses in 2D and 3D. We define $\ell = (\ell_j)_{j=1}^{J-1}$ as the function allowing us to compute the length of the segments of a pose $\mathrm{p}$:

$$\ell_j : \mathrm{p} \mapsto \|\mathrm{p}_j - \mathrm{p}_{\tau(j)}\|_2 \,, \quad 0 < j \leq J-1 \,, \tag{11}$$

where $\tau : \{1, \ldots, J-1\} \to \{0, \ldots, J-1\}$ maps joint indices to the index of their parent joint:

$$\tau(i) = j < i, \quad \text{s.t. } A_{ij} = 1 \,. \tag{12}$$

From assumption A.4, we know that for any pose $\mathrm{p}$ from the training distribution,

$$\forall j, \quad \ell_j(\mathrm{p}) = s_{j,\tau(j)} \,. \tag{13}$$

Given $D = \{(\mathrm{x}_i, \mathrm{p}_i)\}_{i=1}^N \sim \mathrm{P}(\mathrm{x}, \mathrm{p})$, some i.i.d. evaluation data, the MSE of a model $f$ is defined as:

$$\mathrm{MSE}(f; N) = \frac{1}{N} \sum_{i=1}^N \|\mathrm{p}_i - f(\mathrm{x}_i)\|_2^2 \,, \tag{14}$$

and converges to

$$\mathrm{MSE}^*(f) = \mathbb{E}_{\mathrm{x},\mathrm{p}}\big[\|\mathrm{p} - f(\mathrm{x})\|_2^2\big] \tag{15}$$

---

[1] $S^2(0, s_{j,i})$ is homeomorphic to $S^2(\mathrm{p}_{t,j}^G; s_{j,i})$.

as the dataset size $N$ goes to infinity. We then define the oracle MSE minimizer as

$$f^* = \arg\min_f \mathrm{MSE}^*(f)\,. \tag{16}$$

The quantity in (15) is known in statistics as the expected $L_2$-risk and it is a well-known fact that its minimizer is the conditional expectation:

$$f^*(\mathrm{x}) = \mathbb{E}[\mathrm{p}|\mathrm{x} = \mathrm{x}]\,. \tag{17}$$

Thus, since $\ell_j^2$ are strictly convex and $\mathrm{P}(\mathrm{p}|\mathrm{x})$ is non-degenerate according to assumption A.5, we can conclude from Jensen's strict inequality that for all $j$,

$$\ell_j^2(f^*(\mathrm{x})) = \ell_j^2(\mathbb{E}[\mathrm{p}|\mathrm{x} = \mathrm{x}]) < \mathbb{E}[\ell_j^2(\mathrm{p})|\mathrm{x} = \mathrm{x}] = s_{j\tau(j)}^2\,, \tag{18}$$

where the last equality arises from the fact that $\ell_j^2(\mathrm{p})$ is not random according to (13). Thus, given that $\ell_j > 0$ and $s_{j,\tau(j)} > 0$, we can say that $\ell_j(f^*(\mathrm{x})) < s_{j,\tau(j)}$ for all joints $j$. We conclude that the model $f^*$ minimizing $\mathrm{MSE}^*$ predicts poses that violate assumption A.4 and are inconsistent. ∎

As an immediate corollary of proposition 4.2, we may state the following result, which was empirically illustrated in many parts of our paper:

**Corollary B.1.** *Given a fixed training distribution* $\mathrm{P}(\mathrm{x}, \mathrm{p})$ *respecting assumptions A.3-A.5, for all 3D-HPE model $f$ predicting consistent poses,* i.e.*, that respect assumption A.4, there is an inconsistent model $f'$ with lower mean-squared error.*

PROOF. Let $f' \in \arg\min_{\tilde{f}} \mathrm{MSE}^*(\tilde{f})$. According to proposition 4.2, $f'$ is inconsistent. Suppose that the consistent model $f$ is such that

$$\mathrm{MSE}^*(f) \leq \mathrm{MSE}^*(f')\,. \tag{19}$$

Since $\mathrm{MSE}^*$ reaches its minimum at $f'$, we have $\mathrm{MSE}^*(f) = \mathrm{MSE}^*(f')$. Thus, $f \in \arg\min_{\tilde{f}} \mathrm{MSE}^*(\tilde{f})$, which means that $f$ is also inconsistent according to proposition 4.2. That is impossible given that we assumed $f$ to be consistent. We conclude that Eq. (19) is wrong and that

$$\mathrm{MSE}^*(f) > \mathrm{MSE}^*(f')\,. \tag{20}$$

∎

Note that propositions 4.2 and B.1 assume the use of the MSE loss, which is the most widely used loss in 3D-HPE. We can however extend them to the case where MPJPE serves as optimization criteria under an additional technical assumption:

**Corollary B.2.** *The predicted poses minimizing the mean-per-joint-position-error loss are inconsistent if the training poses distribution* $\mathrm{P}(\mathrm{x}, \mathrm{p})$ *verifies Asm. A.3-A.5 and if the joint-wise residuals' norm standard deviation is small compared to the joint-wise loss:*

$$0 \leq j < J\,, \quad \frac{\sqrt{\mathbb{V}_{\mathrm{x},\mathrm{p}}\big[\|\mathrm{p}_j - f_j(\mathrm{x})\|_2\big]}}{\mathbb{E}_{x,\mathrm{p}}\big[\|\mathrm{p}_j - f_j(\mathrm{x})\|_2\big]} \simeq 0\,. \tag{21}$$

PROOF. From proposition 4.2 we know that the poses predicted by the minimizer $f^*$ of

$$\mathrm{MSE}^*(f) = \mathbb{E}_{\mathrm{x},\mathrm{p}}\big[\|\mathrm{p} - f(\mathrm{x})\|_2^2\big] \tag{22}$$

are inconsistent. Let $f_j$ be the component of $f$ corresponding to the $j^{\mathrm{th}}$ joint. We define the $j^{\mathrm{th}}$ mean-per-joint-position-error component as:

$$\mathrm{MPJPE}_j^*(f) \triangleq \mathbb{E}_{\mathrm{x},\mathrm{p}}\big[\|\mathrm{p}_j - f_j(\mathrm{x})\|_2\big]\,. \tag{23}$$

Under the small variance assumption, we have:

$$\frac{\mathbb{V}_{\mathrm{x},\mathrm{p}}\big[\|\mathrm{p}_j - f_j(\mathrm{x})\|_2\big]}{\mathbb{E}_{x,\mathrm{p}}\big[\|\mathrm{p}_j - f_j(\mathrm{x})\|_2\big]^2} \tag{24}$$

$$= \frac{\mathbb{E}_{\mathrm{x},\mathrm{p}}\big[\|\mathrm{p} - f(\mathrm{x})\|_2^2\big] - \mathbb{E}_{\mathrm{x},\mathrm{p}}\big[\|\mathrm{p}_j - f_j(\mathrm{x})\|_2\big]^2}{\mathbb{E}_{x,\mathrm{p}}\big[\|\mathrm{p}_j - f_j(\mathrm{x})\|_2\big]^2} \tag{25}$$

$$= \frac{\mathrm{MSE}_j^*(f) - \mathrm{MPJPE}_j^*(f)^2}{\mathrm{MPJPE}_j^*(f)^2} \simeq 0\,, \tag{26}$$

so both criteria, MSE and MPJPE, are asymptotically equivalent and have the same minimizer $f^*$, which is inconsistent according to proposition 4.2. ∎

**Corollary B.3.** *Under Asm. A.4-A.5 and under (21), the only way to get both optimal MPJPE and consistency is to use multiple hypotheses.*

PROOF. Corollary B.1 and Proposition 4.2 imply that single-hypothesis models (constrained or not) deliver either suboptimal MPJPE or inconsistent pose predictions. Hence, by negation, we get our result. □

In the next section, we further show that multi-hypotheses models, constrained or not, can theoretically show a better $L2$-risk (or *quantization*) performance compared with single-hypotheses models.

## B.2 Multiple hypotheses (constrained or not) can improve L2-risk over single-hypothesis models

Let $\mathcal{X} = \mathbb{R}^{2 \times J}$ denote the space of input 2D poses and $\mathcal{P} = \mathbb{R}^{3 \times J}$ the space of 3D poses. Also, let $\mathcal{R}(f) = \mathbb{E}_{x,p}[\|p - f(x)\|_2^2]$ be the $L2$-risk of some pose estimator $f$ under some underlying continuous joint distribution of 2D-3D pose pairs $P(x, p)$, with density $\rho$ (when it exists).

Before stating the proposition, we need to define an adapted notion of risk for multi-hypothesis models under the oracle aggregation scheme:

**Definition B.4** (Winner-takes-all risk, [41]). As in [41] (section 3.2) and in [23] (section 2.2), we define the $L2$-risk for $K$-head models $f_{\text{WTA}} = (f_{\text{WTA}}^1, \ldots, f_{\text{WTA}}^K)$ as:

$$\mathcal{R}_{\text{WTA}}^K(f_{\text{WTA}}) \triangleq \int_{\mathcal{X}} \sum_{k=1}^{K} \int_{\mathcal{V}^k(f_{\text{WTA}}(x))} \|f_{\text{WTA}}^k(x) - p\|_2^2 \rho(x, p) \, dp \, dx, \tag{27}$$

where $\mathcal{V}^k(g)$'s denotes the $k^{th}$ cell of the Voronoi tesselation of the output space $\mathcal{P}$ defined by generators $g = (g^1, \ldots, g^K) \in \mathcal{P}^K$:

$$\mathcal{V}^k(g) \triangleq \left\{ p \in \mathcal{P} \mid \|g^k - p\|_2^2 < \|g^r, -p\|_2^2, \forall r \neq k \right\}. \tag{28}$$

The risk above translates the notion of oracle pose, since it partitions the space of ground-truth poses $\mathcal{P}$ into regions where some hypothesis is the closest, and uses only that hypothesis to compute the risk in that region. Note that $\mathcal{R}_{\text{WTA}}^1(f) = \mathcal{R}(f)$ for any function $f$, since a single-cell tessellation of $\mathcal{P}$ is $\mathcal{P}$ itself.

In the following, we assume that $f$ is expressive enough, so that, minimizing the risk (27) comes down to minimizing

$$\sum_{k=1}^{K} \int_{\mathcal{V}^k(f_{\text{WTA}}(x))} \|f_{\text{WTA}}^k(x) - p\|_2^2 \rho(x, p) \, dp,$$

for each $x \in \mathcal{X}$.

**Proposition B.5** (Optimality of manifold constrained multi-hypothesis models). *A $K$-hypotheses model $f_{WTA}^* = (f_{WTA}^{1,*}, \ldots, f_{WTA}^{K,*})$ minimizing (27) has always a risk lower or equal to a single-hypothesis model $f_{MSE}^*$ minimizing $\mathcal{R}$:*

$$\mathcal{R}_{WTA}^K(f_{WTA}^*) \leq \mathcal{R}_{WTA}^1(f_{MSE}^*) = \mathcal{R}(f_{MSE}^*). \tag{29}$$

PROOF. Following [23] (Section 2.2), we decouple the cell generators from the risk arguments in (27):

$$\mathcal{K}(g, z) \triangleq \sum_{k=1}^{K} \int_{\mathcal{V}^k(g)} \|z^k - p\|_2^2 \rho(p|x) \, dp, \tag{30}$$

for any generators $g = (g^1, \ldots, g^K) \in \mathcal{P}^K$ and arguments $z = (z^1, \ldots, z^K) \in \mathcal{P}^K$. Note that $\mathcal{R}_{\text{WTA}}^K(f) = \int_{\mathcal{X}} \mathcal{K}(f(x), f(x)) \rho(x) \, dx$.

According to Proposition 3.1 of [7] (or Proposition 2.1 in [23]), if $f_{\text{WTA}}^*$ minimizes $\mathcal{R}_{\text{WTA}}^K$, then $(f_{\text{WTA}}^*(x), f_{\text{WTA}}^*(x))$ has to minimize $\mathcal{K}$ for all $x \in \mathcal{X}$:

$$\mathcal{K}(f_{\text{WTA}}^*(x), f_{\text{WTA}}^*(x)) \leq \mathcal{K}(g, z), \qquad \forall g, z \in \mathcal{P}^K \times \mathcal{P}^K. \tag{31}$$

Let's choose $g$ such that $g^k = f_{\text{WTA}}^{k,*}(x)$ and $z$ such that $z^k = f_{\text{MSE}}^*(x)$ for all $1 \leq k \leq K$. Then

$$\mathcal{R}_{\text{WTA}}^K(f_*^1, \ldots, f_*^K) \leq \int_{\mathcal{X}} \sum_{k=1}^{K} \int_{\mathcal{V}^k(f_{\text{WTA}}^*(x))} \|f_{\text{MSE}}^*(x) - p\|_2^2 \rho(p|x)\rho(x)\,dp\,dx = \mathcal{R}(f_{\text{MSE}}^*), \quad (32)$$

where the last equality comes from the fact that $\mathcal{V}^k(f_{\text{WTA}}^*(x))$ defines a partition of $\mathcal{P}$.

## C  Further details of 1D-to-2D case study

### C.1  Implementation details

**Datasets.** We created a dataset of input-output pairs $\{(x_i, (x_i, y_i))\}_{i=1}^N$, divided into $1\,000$ training examples, $1\,000$ validation examples and $1\,000$ test examples. Since the 2D position of $J_1$ is fully determined by the angle $\theta$ between the segment $(J_0, J_1)$ and the $x$-axis, the dataset is generated by first sampling $\theta$ from a von Mises mixture distribution, then converting it into Cartesian coordinates $(x_i, y_i)$ to form the outputs, and finally projecting them into the $x$-axis to obtain the inputs.

**Distribution scenarios.** We considered three different distribution scenarios with different levels of difficulty:

1. **Easy scenario**: a unimodal distribution centered at $\theta = \frac{2\pi}{5}$, where the axis of maximum 2D variance is approximately parallel to the $x$-axis (Fig. 4-A).

2. **Difficult unimodal scenario**: a unimodal distribution centered at $\theta = 0$, where the axis of maximum 2D variance is perpendicular to the $x$-axis (Fig. 4-B).

3. **Difficult multimodal scenario**: a bimodal distribution, with modes at $\theta_1 = \frac{\pi}{3}$ and $\theta_2 = -\frac{\pi}{3}$ and mixture weights $w_1 = \frac{2}{3}$ and $w_2 = \frac{1}{3}$, *i.e.*, where the projection of modes onto the $x$-axis are close to each other (Fig. 4-C).

All von Mises components in all scenarios had concentrations equal to $20$.

**Architectures and training.** All three models were based on a multi-layer perceptron (MLP) with 2 hidden layers of 32 neurons each, using `tanh` activation.

The constrained and unconstrained MLPs were trained using the mean-squared loss $\frac{1}{N}\sum_{i=1}^N((\hat{x}_i - x_i)^2 + (\hat{y}_i - y_i)^2)$. ManiPose was trained with the loss in Eq. (1), and had $K = 2$ heads. We trained all models with batches of 100 examples for a maximum of 50 epochs. We used the Adam optimizer [17], with default hyperparameters and no weight decay. Learning rates were searched for each model and distribution independently over a small grid: $[10^{-5}, 10^{-4}, 10^{-3}, 10^{-2}]$ (*cf.* selected values in Table 5). They were scheduled during training using a plateau strategy of factor $0.5$, patience of 10 epochs and threshold of $10^{-4}$.

Table 5: **Selected learning rates for 1D-to-2D synthetic experiment.**

| Distribution | A | B | C |
|---|---|---|---|
| Unconstr. MLP | $10^{-3}$ | $10^{-3}$ | $10^{-2}$ |
| Constrained MLP | $10^{-2}$ | $10^{-4}$ | $10^{-2}$ |
| ManiPose | $10^{-2}$ | $10^{-3}$ | $10^{-2}$ |

### C.2  Extension to 2D-to-3D setup with more joints

We further extend the two-joint 1D-to-2D lifting experiment of Section 4.2 to 2D-to-3D with three joints, aiming at providing a scenario that is closer to real-world 3D-HPE, but that can still be fully dissected and visualized.

As in Section 4.2, we suppose that joint $J_0$ is at the origin at all times, that $J_1$ is connected to $J_0$ through a rigid segment of length $s_1$ and that $J_2$ is connected to $J_1$ through a second rigid segment of length $s_1 < s_0$. We further assume that both $J_1$ and $J_2$ are allowed to rotate around two axes

orthogonal to each other. Thus, $J_1$ is constrained to lie on a circle $S^1(0, s_0)$, while $J_2$ lies on a torus $\mathcal{T}$ homeomorphic to $S^1(0, s_0) \otimes S^1(0, s_1)$. Without loss of generality, we set the radii $s_0 = 2$ and $s_1 = 1$ and assume them to be known.

Given that setup, we are interested in learning to predict the 3D pose $(J_1, J_2) = (x_1, y_1, z_1, x_2, y_2, z_2) \in \mathbb{R}^6$, given its 2D projection $(K_1, K_2) = (x_1, z_1, x_2, z_2) \in \mathbb{R}^4$. We create a dataset comprising 20000 training, 2000 validation, and 2000 test examples, sampled using an arbitrary von Mises mixture of poloidal and toroidal angles $(\theta, \phi)$ in $\mathcal{T}$. We set the modes of such a mixture at $[(-\pi, 0), (0, \pi/4), (\frac{1}{2}, -\pi/4), (2\pi/3, \pi/2)]$, with concentrations of $[2, 4, 3, 10]$ and weights $[0.3, 0.4, 0.2, 0.1]$. Similarly to Fig. 4-C, that creates a difficult multimodal distribution, depicted in Fig. 8.

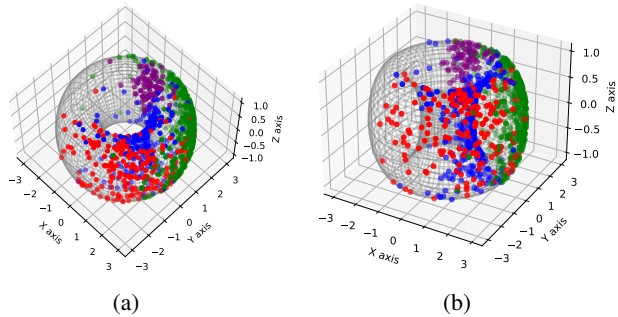

(a)                        (b)

Figure 8: **Visualisation of the von Mises mixture distribution on the torus** $T$**.** The different colors (blue, green, red, purple) represent the modes of the sampled points. We are only representing joint $J_2$ here for clarity.

We train and evaluate the same baselines as in Section 4.2 in that new scenario, using a similar setup (*cf.* Appendix C.1, Architectures and training). Note that for those experiments, we used an initial learning rate of $10^{-3}$ for each baseline, and a batch size of 1000 examples. The corresponding Mean Per Segment Consistency Error (MPSCE) and Mean Per Joint Position Error (MPJPE) results are reported in Table 6.

Table 6: **Mean per joint prediction error (MPJPE) and mean per segment consistency error (MPSCE) in a 2D-to-3D scenario.** Results are averaged over five random seeds. ManiPose reaches perfect MPSCE consistency without degrading MPJPE performance.

| | MPJPE $\downarrow$ | MPSCE $\downarrow$ |
|---|---|---|
| Unconst. MLP | $1.152 \pm 0.021$ | $0.269 \pm 0.018$ |
| Constrained MLP | $1.166 \pm 0.028$ | $\mathbf{0.000 \pm 0.000}$ |
| ManiPose | $\mathbf{1.149 \pm 0.036}$ | $\mathbf{0.000 \pm 0.000}$ |

We see that the same observations as in Section 4.2 also apply here: although the unconstrained MLP yields competitive MPJPE results, its predictions are not consistently aligned with the manifold, as indicated by its poor MPSCE performance. Again, we show here that ManiPose offers an effective balance between maintaining manifold consistency and achieving high joint-position-error performance.

## D   Further ManiPose implementation details

### D.1   Architectural details

Our architecture is backbone-agnostic, as shown on Fig. 2. Thus, in order to have a fair comparison, we decided to implement it using the most powerful architecture available, *i.e.*, MixSTE [52].

In practice, the rotations module follows the MixSTE architecture with $d_l = 8$ spatio-temporal transformer blocks of dimension $d_m = 512$ and time receptive field of $T = 243$ frames for Human 3.6M experiments and $T = 43$ frames for MPI-INF-3DHP experiments. Contrary to MixSTE, that network

outputs rotation embeddings of dimension 6 for each joint and frame, instead of Cartesian coordinates of dimension 3.

Concerning the segment module, it was also implemented with a smaller MixSTE backbone of depth $d_l = 2$ and dimension $d_m = 128$.

The ablation study presented in Table 4 shows that the increase in the number of parameters between MixSTE and ManiPose is negligible.

## D.2 Pose decoding details

The pose decoding block from Fig. 2 is described in Section 3.1 and is based on Algorithms 1 and 2. The whole procedure is illustrated on Fig. 3.

Table 7: Joint-wise weights used in the Winner-takes-all loss Eq. (2) (as in [52]).

| Joint | 0 | 1 | 2 | 3 | 4 | 5 | 6 | 7 | 8 | 9 | 10 | 11 | 12 | 13 | 14 | 15 | 16 |
|---|---|---|---|---|---|---|---|---|---|---|---|---|---|---|---|---|---|
| Weight | 1 | 1 | 2.5 | 2.5 | 1 | 2.5 | 2.5 | 1 | 1 | 1 | 1.5 | 1.5 | 4 | 4 | 1.5 | 4 | 4 |

---

**Algorithm 1** 6D rotation representation conversion [54]

---

**Require:** Predicted 6D rotation representation $r \in \mathbb{R}^6$.
1: $x' \leftarrow [r_0, r_1, r_2]$,
2: $y' \leftarrow [r_3, r_4, r_5]$,
3: $x \leftarrow x'/\|x'\|_2$,
4: $z' \leftarrow x \wedge y'$,
5: $z \leftarrow z'/\|z'\|_2$,
6: $y \leftarrow z \wedge x$,
7: **return** $R = [x|y|z] \in \mathbb{R}^{3 \times 3}$.

---

**Algorithm 2** Forward Kinematics [34, 26]

---

**Require:** Scaled reference pose $u' \in (\mathbb{R}^3)^J$, predicted rotation matrices $R_{t,j}, 0 \le j < J$.
1: $R'_{t,0} \leftarrow R_{t,0}$,
2: $\mathrm{p}_{t,0} \leftarrow \mathrm{u}'_0$,
3: **for** $j = 1, \ldots, J - 1$ **do**
4:      $R'_{t,j} \leftarrow R_{t,j} R'_{t,\tau(j)}$,                     ▷ Compose relative rotations
5:      $\mathrm{p}_{t,j} \leftarrow R'_{t,j}(\mathrm{u}'_j - \mathrm{u}'_{\tau(j)}) + \mathrm{p}_{t,\tau(j)}$,
6: **end for**
7: **return** $\mathrm{p}_t = [\mathrm{p}_{t,j}]_{0 \le j < J}$

---

## D.3 Training details

**Training tactics.** In order to have a fair comparison with MixSTE [52], we trained ManiPose using the same training tactics, such as pose flip augmentation both at training and test time. Moreover, the training loss (1) was complemented with two additional terms described in [52]:

1. a TCloss term, initially introduced in [13];
2. a velocity loss term, introduced in [38].

We also weighted the Winner-takes-all MPJPE loss (2) as in [52] (*cf.* weights in Table 7). The score loss weight, $\beta$, was set to 0.1 according to our hyperparameter study (Appendix E), while TCloss and velocity loss terms had respective weights of 0.5 and 2 (values from [52]).

**Training settings.** We trained our model for a maximum of 200 epochs with the Adam optimizer [17], using default hyperparameters, a weight decay of $10^{-6}$ and an initial learning rate of $4 \times 10^{-5}$. The latter was reduced with a plateau scheduler of factor 0.5, patience of 11 epochs and threshold of 0.1 mm. Batches contained 3 sequences of $T = 243$ frames each for the Human 3.6M training, and 30 sequences of $T = 43$ frames for MPI-INF-3DHP.

**Compute resources.** Trainings were carried out on a single NVIDIA RTX 2000 GPU with around 11GB of memory. The training of the large model with 243 frames on Human 3.6M dataset took around 26 hours.

**Dataset licences.** Human 3.6M is a dataset released under a research-only custom license, and is available upon request at this URL: http://vision.imar.ro/human3.6m/description.php. MPI-INF-3DHP is released under non-commercial custom license and can be found at: https://vcai.mpi-inf.mpg.de/3dhp-dataset/.

## D.4 Baselines evaluation.

All Human 3.6M evaluations of MPSSE and MPSCE listed in Tables 2 and 4 were performed using the official checkpoints of these methods and their corresponding official evaluation scripts. Concerning MPI-INF-3DHP evaluations from Table 3, checkpoints were not available (except for P-STMO). Thus, baseline models were retrained from scratch using the official MPI-INF-3DHP training scripts provided by the authors of each work, using hyperparameters reported in their corresponding papers. We checked that we were able to reproduce the reported MPJPE results.

# E   Further results on the Human 3.6M dataset

Table 8: **Quantitative comparison with the state-of-the-art methods on Human3.6M under Protocol #1 (MPJPE in mm), using detected 2D poses.** $T$: sequence length. $K$: number of hypotheses. Orac.: Metric computed using oracle hypothesis. **Bold**: best; Underlined: second best.

| | $T$ | $K$ | Orac. | Dir. | Disc | Eat | Greet | Phone | Photo | Pose | Purch. | Sit | SitD. | Smoke | Wait | WalkD. | Walk | WalkT. | Avg. |
|---|---|---|---|---|---|---|---|---|---|---|---|---|---|---|---|---|---|---|---|
| *Single-hypothesis methods:* | | | | | | | | | | | | | | | | | | | |
| GraphSH [51] | 1 | 1 | | 45.2 | 49.9 | 47.5 | 50.9 | 54.9 | 66.1 | 48.5 | 46.3 | 59.7 | 71.5 | 51.4 | 48.6 | 53.9 | 39.9 | 44.1 | 51.9 |
| MGCN [55] | 1 | 1 | | 45.4 | 49.2 | 45.7 | 49.4 | 50.4 | 58.2 | 47.9 | 46.0 | 57.5 | 63.0 | 49.7 | 46.6 | 52.2 | 38.9 | 40.8 | 49.4 |
| ST-GCN [2] | 7 | 1 | | 44.6 | 47.4 | 45.6 | 48.8 | 50.8 | 59.0 | 47.2 | 43.9 | 57.9 | 61.9 | 49.7 | 46.6 | 51.3 | 37.1 | 39.4 | 48.8 |
| VideoPose3D [38] | 243 | 1 | | 45.2 | 46.7 | 43.3 | 45.6 | 48.1 | 55.1 | 44.6 | 44.3 | 57.3 | 65.8 | 47.1 | 44.0 | 49.0 | 32.8 | 33.9 | 46.8 |
| UGCN [47] | 96 | 1 | | 41.3 | 43.9 | 44.0 | 42.2 | 48.0 | 57.1 | 42.2 | 43.2 | 57.3 | 61.3 | 47.0 | 43.5 | 47.0 | 32.6 | 31.8 | 45.6 |
| Liu *et al.* [28] | 243 | 1 | | 41.8 | 44.8 | 41.1 | 44.9 | 47.4 | 54.1 | 43.4 | 42.2 | 56.2 | 63.6 | 45.3 | 43.5 | 45.3 | 31.3 | 32.2 | 45.1 |
| PoseFormer [53] | 81 | 1 | | 41.5 | 44.8 | 39.8 | 42.5 | 46.5 | 51.6 | 42.1 | 42.0 | 53.3 | 60.7 | 45.5 | 43.3 | 46.1 | 31.8 | 32.2 | 44.3 |
| Anatomy3D [4] | 243 | 1 | | 41.4 | 43.2 | 40.1 | 42.9 | 46.6 | 51.9 | 41.7 | 42.3 | 53.9 | 60.2 | 45.4 | 41.7 | 46.0 | 31.5 | 32.7 | 44.1 |
| MixSTE [52] | 243 | 1 | | 37.6 | 40.9 | 37.3 | 39.7 | 42.3 | 49.9 | 40.1 | 39.8 | 51.7 | 55.0 | 42.1 | 39.8 | 41.0 | 27.9 | 27.9 | 40.9 |
| *Multi-hypothesis methods:* | | | | | | | | | | | | | | | | | | | |
| Li *et al.* [25] | 1 | 10 | ✓ | 62.0 | 69.7 | 64.3 | 73.6 | 75.1 | 84.8 | 68.7 | 75.0 | 81.2 | 104.3 | 70.2 | 72.0 | 75.0 | 67.0 | 69.0 | 73.9 |
| Li *et al.* [24] | 1 | 5 | ✓ | 43.8 | 48.6 | 49.1 | 49.8 | 57.6 | 61.5 | 45.9 | 48.3 | 62.0 | 73.4 | 54.8 | 50.6 | 56.0 | 43.4 | 45.5 | 52.7 |
| Oikarinen *et al.* [36] | 1 | 200 | ✓ | 40.0 | 43.2 | 41.0 | 43.4 | 50.0 | 53.6 | 40.1 | 41.4 | 52.6 | 67.3 | 48.1 | 44.2 | 44.9 | 39.5 | 40.2 | 46.2 |
| Sharma *et al.* [44] | 1 | 10 | ✓ | 37.8 | 43.2 | 43.0 | 44.3 | 51.1 | 57.0 | 39.7 | 43.0 | 56.3 | 64.0 | 48.1 | 45.4 | 50.4 | 37.9 | 39.9 | 46.8 |
| Wehrbein *et al.* [49] | 1 | 200 | ✓ | 38.5 | 42.5 | 39.9 | 41.7 | 46.5 | 51.6 | 39.9 | 40.8 | 49.5 | 56.8 | 45.3 | 46.4 | 46.8 | 37.8 | 40.4 | 44.3 |
| DiffPose [12] | 1 | 200 | ✓ | 38.1 | 43.1 | **35.3** | 43.1 | 46.6 | 48.2 | 39.0 | 37.6 | 51.9 | 59.3 | 41.7 | 47.6 | 45.4 | 37.4 | 36.0 | 43.3 |
| MHFormer [27] | 351 | 3 | | 39.2 | 43.1 | 40.1 | 40.9 | 44.9 | 51.2 | 40.6 | 41.3 | 53.5 | 60.3 | 43.7 | 41.1 | 43.8 | 29.8 | 30.6 | 43.0 |
| D3DP [43] | 243 | 20 | ✓ | 37.3 | 39.4 | 35.4 | 37.8 | 41.3 | 48.1 | 39.0 | 37.9 | 49.8 | 52.8 | 41.1 | 39.0 | 39.4 | 27.3 | **27.2** | 39.5 |
| ManiPose (Ours) | 243 | 5 | ✗ | 39.6 | 45.8 | 41.9 | 37.1 | 42.7 | 52.3 | 47.7 | 39.5 | 42.7 | 53.3 | 42.6 | 40.9 | 48.2 | 27.0 | 30.0 | 42.1 |
| ManiPose (Ours) | 243 | 5 | ✓ | **36.0** | 41.5 | 38.9 | **34.5** | 39.6 | 48.5 | 42.7 | **37.4** | 39.8 | 50.0 | 40.2 | 37.7 | 45.3 | **25.9** | 28.6 | **39.1** |

**Protocol #1 and #2 detailed results.** A detailed quantitative comparison in terms of MPJPE per action on Human3.6M dataset between ManiPose and state-of-the-art methods is shown in Table 8. We see that ManiPose reaches the best MPJPE performance on average and on most actions. Table 9 contains a similar analysis in terms of P-MPJPE (*i.e.*, MPJPE with procrust-aligned poses). We observe the same patterns as in Table 8, namely that ManiPose reaches the second-best P-MPJPE performance on average and for most actions. We confirm here again that the substantial improvements in pose consistency brought by ManiPose are not obtained at the expense of traditional metrics derived from MPJPE.

**Errors per joint.** On the top of Fig. 9 we see that most of MixSTE errors come from feet, elbows and wrist joints, which are most prone to depth ambiguity. ManiPose helps to reduce the position errors for most of those ambiguous joints, probably as a byproduct of its major consistency improvements shown in Fig. 5.

**Impact of hyperparameters.** ManiPose introduces two additional hyperparameters when compared to MixSTE: the number $K$ of hypotheses and the score loss weight $\beta$ (*cf.* Eq. (1)). We further assess the impact of their respective values on MPJPE. For computational cost reasons, we used a smaller version of our model for this study, with transformer blocks of dimension $d_m = 64$ and time receptive field of $T = 27$ frames. Fig. 10 (left) shows that more hypotheses help, but that the performance

Table 9: **Quantitative comparison with the state-of-the-art methods on Human3.6M under Protocol #2 (P-MPJPE in mm), using detected 2D poses. Bold**: best; Underlined: second best. ManiPose results using the oracle evaluation. Actions: `Directions, Discussion, Eating, Greeting, Talking on the Phone, Taking photo, Posing, Makes purchases, Sitting on chair, Activities while seated, Smoking, Waiting, Walking dog, Walking, Walking together.`

| | $T$ | $K$ | Dir. | Disc | Eat | Greet | Phone | Photo | Pose | Purch. | Sit | SitD. | Smoke | Wait | WalkD. | Walk | WalkT. | Avg. |
|---|---|---|---|---|---|---|---|---|---|---|---|---|---|---|---|---|---|---|
| MGCN [55] | 1 | 1 | 35.7 | 38.6 | 36.3 | 40.5 | 39.2 | 44.5 | 37.0 | 35.4 | 46.4 | 51.2 | 40.5 | 35.6 | 41.7 | 30.7 | 33.9 | 39.1 |
| ST-GCN [2] | 1 | 1 | 35.7 | 37.8 | 36.9 | 40.7 | 39.6 | 45.2 | 37.4 | 34.5 | 46.9 | 50.1 | 40.5 | 36.1 | 41.0 | 29.6 | 33.2 | 39.0 |
| Pavllo *et al.* [38] | 243 | 1 | 34.2 | 36.8 | 33.9 | 37.5 | 37.1 | 43.2 | 34.4 | 33.5 | 45.3 | 52.7 | 37.7 | 34.1 | 38.0 | 25.8 | 27.7 | 36.8 |
| Zheng *et al.* [53] | 81 | 1 | 34.1 | 36.1 | 34.4 | 37.2 | 36.4 | 42.2 | 34.4 | 33.6 | 45.0 | 52.5 | 37.4 | 33.8 | 37.8 | 25.6 | 27.3 | 36.5 |
| Liu *et al.* [28] | 243 | 1 | 32.3 | 35.2 | 33.3 | 35.8 | 35.9 | 41.5 | 33.2 | 32.7 | 44.6 | 50.9 | 37.0 | 32.4 | 37.0 | 25.2 | 27.2 | 35.6 |
| Anatomy3D [4] | 243 | 1 | 32.6 | 35.1 | 32.8 | 35.4 | 36.3 | 40.4 | 32.4 | 32.3 | 42.7 | 49.0 | 36.8 | 32.4 | 36.0 | 24.9 | 26.5 | 35.0 |
| UGCN [47] | 96 | 1 | 31.8 | 34.3 | 35.4 | 33.5 | 35.4 | 41.7 | **31.1** | 31.6 | 44.4 | 49.0 | 36.4 | 32.2 | 35.0 | 24.9 | 23.0 | 34.5 |
| MixSTE [52] | 243 | 1 | **30.8** | **33.1** | **30.3** | **31.8** | **33.1** | **39.1** | **31.1** | **30.5** | 42.5 | **44.5** | **34.0** | **30.8** | **32.7** | **22.1** | **22.9** | **32.6** |
| ManiPose (Ours) | 243 | 5 | 31.9 | 35.7 | 30.8 | 33.5 | 34.0 | 39.8 | 33.0 | 31.4 | **41.1** | 45.9 | 36.0 | 32.3 | 35.4 | 24.7 | 25.8 | 34.1 |

improvements saturate around 5 hypotheses. Concerning $\beta$, Fig. 10 (right) shows that lower values help to improve the MPJPE performance.

**Impact of the rotations representations used.**

The disentanglement between segments' length and orientation is not novel, and was proposed in previous works restricted to the single-hypothesis case, such as Anatomy3D [4]. While ManiPose represents segments' orientations as full 3D rotations relatively to parent segments in the kinematics tree, Anatomy3D simply predicts segments' absolute directions, *i.e.*, normalized vectors in the 3D space. This solution has the advantage of not over-parametrizing the segments orientations (which are invariant to rotations around the segment axis) and being lower dimensional (3 vs 6). One might hence wonder whether Anatomy3D's parametrization is not preferable. As shown in Table 10, Anatomy3D's implementation led to poorer results when compared to our rotations parametrization in a multi-hypothesis setting. This motivated us to use full 3D rotations' representations proposed in [54] in our experiments, despite their caveats. Note that [54] also shows good empirical results in the related problem of inverse kinematics of human 3D poses.

Table 10: **Rotations representation ablation: learning 3D directions instead of full rotations yields poorer results.** Dim.: Dimension of rotations or directions representations. $K$: Number of hypotheses. $\beta$ Scores regularization. **Bold**: best. Underlined: second best.

| | Learn | Dim. | $K$ | $\beta$ | MPJPE↓ | MPSSE↓ | MPSCE↓ |
|---|---|---|---|---|---|---|---|
| ManiPose (Ours) | Rotations | 6 | 5 | 0.1 | **39.1** | **0.3** | **0.5** |
| Anatomy3D-like [4] | Directions | 3 | 5 | 0.1 | 39.6 | 3.2 | 5.9 |
| | Directions | 3 | 5 | 0.5 | 41.8 | 3.9 | 6.9 |
| | Directions | 3 | 3 | 0.5 | 43.2 | 4.4 | 7.5 |

**Diversity of predicted poses.** As explained in Section 5.2, ManiPose's state-of-the-art oracle MPJPE results show that it excels in terms of diversity when the latter is assessed using the quantization error. There are many other ways of assessing distribution diversity. In an attempt to quantify the diversity of pose distributions learned by ManiPose by other means, we have computed the coverage (as defined in [35]) of generated poses relatively to the ground-truth test set of Human 3.6M. For computation cost reasons (it grows quadratically with sample size), we limited our analysis to 5 actions from subject S11. We compare ManiPose to DiffPose [10], using 5 hypotheses for both, and observe similar diversity on average (*cf.* Fig. 11).

# F  Code

We provide the code to reproduce all our experiments under https://github.com/cedricrommel/manipose.

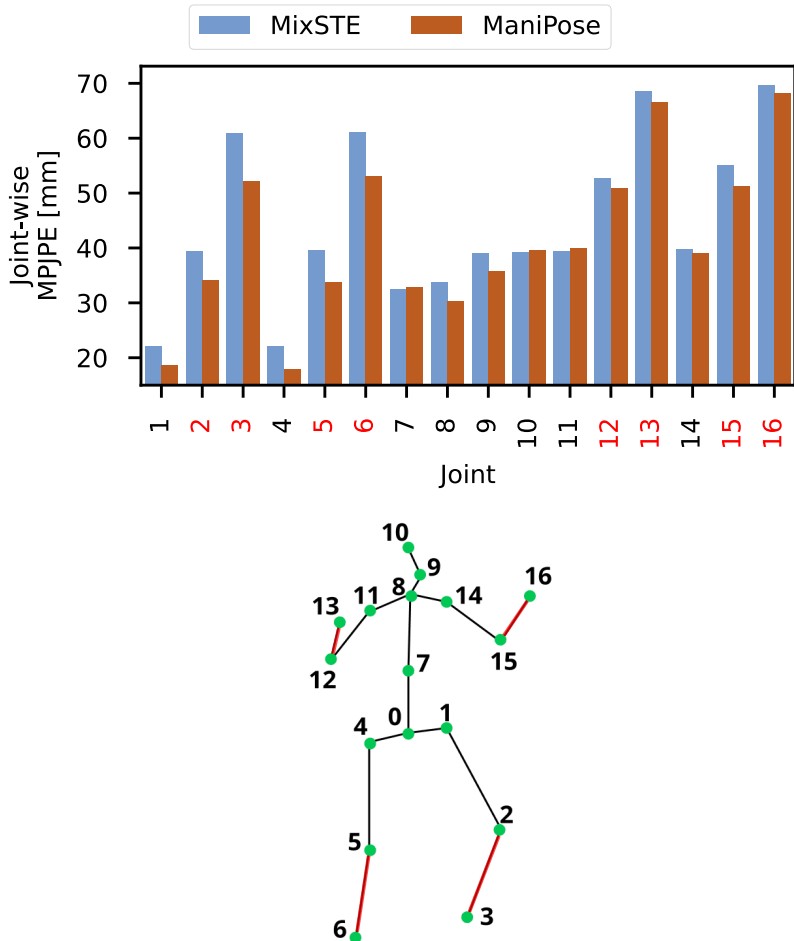

Figure 9: Detailed results on H3.6M. **Top:** Mean position errors per joint. **Bottom:** Human 3.6M skeleton.

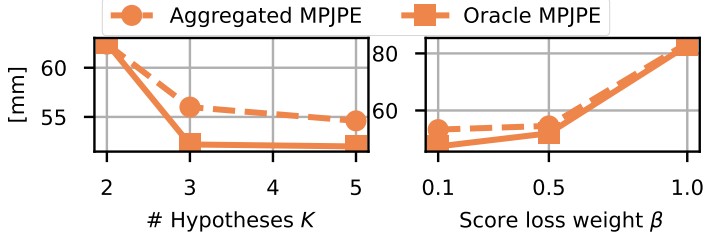

Figure 10: **Impact of the number $K$ of hypotheses (left) and score loss weight $\beta$ (right) on ManiPose aggregated and oracle performance.** Results are obtained on H3.6M with a smaller network ($d_m = 64$) and a shorter sequence ($T = 27$). Left plot obtained with $\beta = 0.1$ and right plot with $K = 5$.

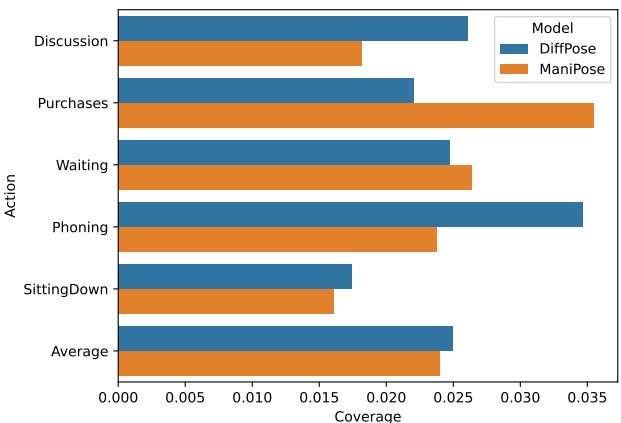

Figure 11: **ManiPose achieves similar diversity to DiffPose [10].** Diversity is assessed through the coverage [35] over test data from subject 11 from Human 3.6M. 5 hypotheses were predicted/sampled for each frame by both models.

