# OpenReview forum: "ManiPose: Manifold-Constrained Multi-Hypothesis 3D Human Pose Estimation"
_NeurIPS.cc/2024/Conference — NeurIPS 2024 poster_

### Official Review · Reviewer_fpqy · 2024-07-09

**Soundness:** 2
**Presentation:** 3
**Contribution:** 1
**Rating:** 3
**Confidence:** 4

**Summary:**

This paper presents a method to estimate 3D human keypoints from a sequence of monocular 2D keypoints observations. It builds upon an existing sequence-to-sequence architecture (MixSTE), with a different output parameterization exploiting a kinematic skeletton prior, and different training losses. Lengths of the skeletton bones are predicted for the whole sequence to ensure consistency across frames (and maybe also left/right symmetry of the skeletton), and five 3D pose hypotheses with associated scores are predicted for each frame, parameterized as a list of 3D relative orientation for each bone with respect to its parent in the kinematic tree.

The authors develop theoretical arguments regarding the benefits of enforcing such structural priors in the predictions, and illustrate with a toy example the interest of having multiple predictions in case of ambiguous multimodal output. They validate their approach on Human3.6M and MPI-INF-3DHP datasets.

**Strengths:**

The motivation for exploiting bone lengths constraints is well expressed, with a clear and detailed discussion provided in Section 4. The discussion of experimental and ablation results is insightful and shows – in a setting dependent on an oracle – benefits of the proposed approach.

**Weaknesses:**

The idea of enforcing body priors (constant bone length here) is not novel and has actually been heavily exploited in a whole line of work relying on more advanced parametric models such as SMPL [100]. This line of work would deserve being considered in the paper, as it encompass approaches suitable for 2D-to-3D sequence lifting such as e.g. [101].

The authors present a pose space consisting in 3D coordinates of joints linked by some rigid segments. Based on this definition, a natural pose parameterization would consist in the 3D direction of each segment, yet the authors chose to overparameterize poses by using relative 3D bone orientation instead. I understand that such choice can have practical benefits in term of biomechanical constraints and additional supervision signal when ground truth data is available, but such choice should be properly motivated, discussed and ablated in the paper.


The authors describe two ways of aggregating results L247 but do not state which one they use for MPI-INF-3DHP, and they only report oracle results on Human3.6M and for the ablations.

In my understanding, pose hypotheses are selected independently for each frame and there are no temporal terms in the training objectives or aggregation method. Since the proposed approach deals with temporal sequences, it would be worth evaluating the temporal consistency of the predictions, through qualitative video examples and quantitatively e.g. using joint acceleration metrics. Having multiple hypotheses for each frame brings combinatorial questions worth discussing in my opinion.

References:
- [100] Loper at al., “SMPL: A Skinned Multi-Person Linear Model”, at SIGGRAPH Asia 2015.
- [101] Baradel et al., “PoseBERT: A Generic Transformer Module for Temporal 3D Human Modeling”, in TPAMI 2022.

**Questions:**

See the weaknesses section for a list of suggestions.

---

> ### Author Rebuttal · Authors · 2024-08-07
>
> We thank the reviewer for their comments. We answer their remarks below, following the same order.
>
> 1. **Citation of SMPL-based methods:**
>
> - _We will properly cite these important works._ We agree with the reviewer that SMPL-based methods share the same constant-bone-length assumption that we present, and that it represents an extensive line of work which is worth mentioning. We thank the reviewer for pointing this out to us.
>
> - We would like to highlight however a few differences between these works and ours:
>
>     - _HMR is a field solving a more complex task:_ “Human body pose and shape estimation” (a.k.a. “human body reconstruction”, “human mesh reconstruction”, “HMR”, …) is a separate field, different from our “3D human pose lifting”: the objective of the former is to predict whole 3D body meshes from images, as opposed to 3D joint positions from 2D keypoints for the latter. This means that these works tackle a different task, which is more challenging.
>     - _HMR is often frame-based:_ Because their task is more challenging, SMPL methods are more computation-heavy and often restricted to single-frame predictions, which are then smoothed across frames using optimization-based post-processing. A good example of this methodology can be seen in [A].
>     - _SMLP methods deliver worse MPJPE:_ Of course, predicting the human mesh across time includes the skeleton pose, which means that some of these methods do evaluate MPJPE. Note however that, while [A] is SOTA in Human3.6M data last time we checked, their performance (MPJPE=44.8 mm) lags behind 3D human pose lifting methods (c.f. Table 2 and 8 in our paper).
>     - _Our novelty lies on multiple hypotheses with constraints:_ Furthermore, please note that our work proposes not just to restrict predicted poses to have constant bone length, but proves that this is not sufficient to optimize both pose fitting (i.e., joint position error) and consistency (i.e., avoiding bone stretching). We prove that multiple hypotheses are needed to accommodate both objectives and propose a practical implementation, show-cased in our experimental results.
>
> 2. **Concerns regarding rotations parametrization:**
>
> - _Learning 3D directions instead of full rotations yielded poorer results:_ We understand the reviewer’s concern. It is true that, when compared to full 3D rotation matrices, predicting simple direction vectors (as done in Anatomy3D [4] for example) is a simpler way of parametrizing poses where joints are connected by segments. As advised by the reviewer, we hence performed a new ablation study, where we compare our parametrization to the one used in Anatomy3D, i.e., where the rotations module predicts normalized vectors representing the direction of each segment. As shown in Table 1 of the pdf attached to our global answer, the latter leads to poorer oracle MPJPE of 39.6 mm when compared to our rotations parametrization (39.1 mm) in a multi-hypothesis setting.
>
> - _Our choice was guided by relevant previous works:_ To clarify our choice, we opted for our rotations representation building on results from [43], where the benefits of 6D representations of SO(3) matrices is studied. Please note that [43] includes good results in an inverse kinematics problem with "stick" human poses similar to ours.
>
> 3. **Concern regarding results with aggregated poses:**
>
> - _Values for MPI-INF-3DHP are reported in Table 4:_ The penultimate row corresponds to a weighted averaged pose, while the last row corresponds to the oracle pose.
>
> - _Values for Human3.6M are reported in Table 8 in the appendix:_ Metrics of the aggregated pose can be found at the penultimate row of Table 8. We understand that this can be missed since it is in the appendix.
>
> - _We will clarify this in our revision._ We apologize if the results with aggregated poses are not stated with enough clarity in our manuscript.
>
> - _MPSCE performance is reported above:_ As explained to reviewer 5BLY in Q3, both MPJPE and MPSCE performance are hurt when using the averaged pose. While the latter is expected according to Proposition 4.2 in the paper, the MPJPE degradation is explained by a new theoretical result proved in the answer to reviewer 5BLY.
>
> 4. **Concern regarding temporal consistency of predictions:**
> - _The training objective includes a temporal term in the Human 3.6M and MPI-INF-3DHP experiments._ We apologize if this was not very clear in our manuscript. It is mentioned in Appendix D.3 lines 602-609, where we explain that TCloss and velocity loss terms were added to our original objective in order to have a fair comparison with existing art. We chose not to add this in the main manuscript because of space limitations and because they are relatively standard practice. We agree that it may lead to confusion and will hence add a longer note on this in our revision to the main paper.
>
> - _We provide a gif demonstrating temporal consistency:_ Although we understand the reviewers concern, we found no important issues regarding this point. We believe this to be explained by the aforementioned loss terms and the fact that the model predicts sequences instead of single frames. The reviewer can find a gif in the pdf attached to the main response (NeurIPS rules forbid us from providing a link to a video). Note that Adobe Reader is needed to visualize it.
>
> [A] Goel, Shubham, et al. "Humans in 4D: Reconstructing and tracking humans with transformers." Proceedings of the IEEE/CVF International Conference on Computer Vision. 2023.
>
> [4] Chen, Tianlang, et al. "Anatomy-aware 3d human pose estimation with bone-based pose decomposition." IEEE Transactions on Circuits and Systems for Video Technology 32.1 (2021): 198-209.
>
> [43] Zhou, Yi, et al. "On the continuity of rotation representations in neural networks." Proceedings of the IEEE/CVF conference on computer vision and pattern recognition. 2019.

---

> > ### Comment · Reviewer_fpqy · 2024-08-13
> >
> > Thank you for your answer. I acknowledge having read the rebuttal (but I don't have Acrobat Reader unfortunately).
> >
> > For reference, and although this is not directly related to your argument, the continuity theory proposed in [43] is disputed in:
> > Brégier, "Deep Regression on Manifolds: A 3D Rotation Case Study", in 3DV 2021.

---

### Official Review · Reviewer_5BLY · 2024-07-11

**Soundness:** 3
**Presentation:** 4
**Contribution:** 4
**Rating:** 8
**Confidence:** 4

**Summary:**

This paper proposes a MCL-based framework for multi-hypothesis 3D human pose estimation. This framework predicts skeletal parameters so that the predicted 3D poses in a sequence are constrained to one smooth manifold. To prove the superiority of such a framework, the paper presents detailed theoretical analysis on the drawback of unconstrained single-hypothesis HPE and why MPJPE alone is not enough for pose evaluation. The experiments show the proposed framework is capable of keeping the consistency of predicted poses and achieving state-of-the-art MPJPE in the meantime.

**Strengths:**

* Simple and reasonable manifold representation. The proposed framework keeps the predicted human pose on the target manifold by representing the human pose with bone lengths and orientations, and the 3D pose is a direct inference from forward kinematics. The manifold is represented by the kinematics itself.

* Inspiring theoretical analysis on basic problems in 3D HPE. The paper arrives at some theoretical conclusions (line178-183), along with detailed proofs. They can provide some refreshing ideas on the innate drawbacks of traditional loss functions and MPJPE metrics.

* Good performance under both MPJPE and consistency measures, as validated in Table 2 and 3.

**Weaknesses:**

* Theoretical analysis on the advantage of multi-hypothesis methods over single-hypothesis ones could be added. Specifically, why a **constrained multi-hypothesis** method performs better than an **unconstrained single-hypothesis** method in MPJPE? Though this is already validated by the experiments, I personally believe it would make the paper more solid if the authors could make this analysis.

Minor problem:
* In Fig.4 (C) and (D), it is not quite clear how the estimations (crosses and triangles) correspond with the inputs (black dots). There might be some unexpected shifts, as the projections of the predicitons do not strictly align with the inputs (like in B).

**Questions:**

What is the quality of the score for each hypothesis? If the multiple hypotheses are fused to one (e.g. by taking the one with the largest confidence or taking the weighted average), then how will the MPJPE, MPSCE, and MPSSE change?

**Limitations:**

Yes.

---

> ### Author Rebuttal · Authors · 2024-08-07
>
> We thank the reviewer for their comments. We answer their remarks below, following the same order.
>
> 1. **New theoretical analysis on the advantage of multi-hypothesis methods over single-hypothesis:**
>
> We agree with the reviewer and provide the proposed theoretical result hereafter.
>
> - Let $\mathcal{X}=\mathbb{R}^{2 \times J}$ denote the space of input 2D poses and $\mathcal{P}=\mathbb{R}^{3 \times J}$ the space of 3D poses. Following Rupprecht et al. 2017 [31] and Letzelter et al. 2024 [A], we define the “oracle risk” for a K-hypothesis model $f_{\text{WTA}} = (f_{\text{WTA}}^1, \dots, f_{\text{WTA}}^K)$ as:
> $$  \mathcal{R}^K (f_{\text{WTA}}) \triangleq \int_\mathcal{X}
> \sum_{k=1}^{K} \int_{\mathcal{V}^k(f_\text{WTA}(\mathrm{x}))} \\|f_{\text{WTA}}^k(\mathrm{x}) - \mathrm{p} \\|^2_2 \rho(\mathrm{x}, \mathrm{p}) \mathrm{d} \mathrm{p} \mathrm{d} \mathrm{x},  $$
> where $\mathcal{V}^k(g)$ denotes the $k^{th}$ cell of the Voronoi tesselation of the output space $\mathcal{P}$ defined by generators $g=(g^1, \dots, g^K) \in \mathcal{P}^K$:
> $$\mathcal{V}^k(g) \triangleq \Big\\{ \mathrm{p} \in \mathcal{P} \;\Big|\; \\| g^k - \mathrm{p} \\|^2_2 < \\| g^r - \mathrm{p} \\|^2_2, \forall r \neq k \Big\\}.$$
> The risk above translates the notion of oracle pose, since it partitions the space of ground-truth poses $\mathcal{P}$ into regions where some hypothesis is the closest, and uses only that hypothesis to compute the risk in that region.
>
> _Now we can state our new proposition:_
>
> - A $K$-hypothesis model $f_{\text{WTA}}^*=(f_{\text{WTA}}^{1*}, \dots, f_{\text{WTA}}^{K*} )$ minimizing $\mathcal{R}^K$ has always a risk lower or equal to a single-hypothesis model $f_\text{MSE}^*$ minimizing $\mathcal{R}^1$:
> $$\mathcal{R}^K (f_{\text{WTA}}^*) \leq \mathcal{R}^1(f_\text{MSE}^*) = \min_f \mathbb{E}_{\mathrm{x}, \mathrm{p}}[\\| \mathrm{p} - f(\mathrm{x})\\|^2_2]  = \mathbb{E}\_{\mathrm{x}, \mathrm{p}} [ \mathrm{p}  \| \mathrm{x} ].$$
>
> _The proof relies on the following steps:_
>
> - First we assume that $f_{\text{WTA}}$ is expressive enough, so that, minimizing the risk $\mathcal{R}^K$ comes down to minimizing
> $$\mathcal{R}\_{\mathrm{x}}^K (f\_{\text{WTA}}) \triangleq \sum_{k=1}^{K} \int_{\mathcal{V}^k(f_\text{WTA}(\mathrm{x}))} \\|f_{\text{WTA}}^k(\mathrm{x}) - \mathrm{p} \\|^2_2 \rho(\mathrm{p} | \mathrm{x}) \mathrm{d} \mathrm{p} ,$$
> for each $\mathrm{x} \in \mathcal{X}$.
>
> - Following [A] (Section 2.2), we decouple the cell generators from the risk arguments:
> $$\mathcal{K}(g, z) \triangleq \sum_{k=1}^{K} \int_{\mathcal{V}^k(g)} \\|z^k - \mathrm{p} \\|^2_2 \rho(\mathrm{p} | \mathrm{x}) \mathrm{d} \mathrm{p},$$
> for any generators $g=(g^1, \dots, g^K) \in \mathcal{P}^K$ and arguments $z=(z^1, \dots, z^K) \in \mathcal{P}^K$.
> Note that $\mathcal{R}_{\mathrm{x}}^K(f) = \mathcal{K}(f(\mathrm{x}), f(\mathrm{x}))$.
>
> - Next, according to Proposition 3.1 of [B], if $f_{\text{WTA}}^*$ minimizes the input-dependent risk $\mathcal{R}\_{\mathrm{x}}^K (f\_{\text{WTA}})$, then $(f_{\text{WTA}}^*(\mathrm{x}), f_{\text{WTA}}^*(\mathrm{x}))$ has to minimize $\mathcal{K}$:
> $$\mathcal{K}(f_{\text{WTA}}^*(\mathrm{x}), f_{\text{WTA}}^*(\mathrm{x})) \leq \mathcal{K}(g, z), \qquad \forall g, z \in \mathcal{P}^K \times \mathcal{P}^K.$$
>
> - Finally, let's choose $g$ such that $g^k=f_{\text{WTA}}^{k*}(\mathrm{x})$ and $z$ such that $z^k = f_\text{MSE}^*(\mathrm{x})$ for all $1 \leq k \leq K$. Then
> $$\mathcal{R}\_\\mathrm{x}^K (f\_{\text{WTA}}^{k*}) \leq
> \sum_{k=1}^{K} \int_{\mathcal{V}^k(f_\text{WTA}^*(\mathrm{x}))} \\|f_\text{MSE}^*(\mathrm{x}) - \mathrm{p} \\|^2_2 \rho(\mathrm{p} | \mathrm{x}) \mathrm{d} \mathrm{p} = \mathcal{R}^1_\mathrm{x}(f_\text{MSE}^*) ,$$
> where the last equality comes from the fact that $\mathcal{V}^k(f_\text{WTA}^*(\mathrm{x}))$ defines a partition of $\mathcal{P}$.
>
> 2. **Fig.4 misalignments:**
>
> Indeed, it seems that some misalignment was introduced during editing and should be corrected in our revision.
>
> 3. **Metrics with hypotheses fused to one:**
>
> - _MPJPE degrades:_ When we compute the weighted average (cf. penultimate row in tables 3 and 8 in the paper), we see that MPJPE performance is hurt (42.1 mm in H3.6M instead of 39.1). This might indeed indicate that scores could be better estimated, but it could also just be a consequence of the new proposition proved above in point 1.
>
> - _Expected from our new theoretical result:_ Indeed, according to [17, Equation 8], the weighted average is an estimate of the conditional expectation $\mathbb{E}[\mathrm{p} \| \mathrm{x}]$, i.e., the best single-hypothesis model $f\_\text{MSE}^*$. Hence, according to the proposition proven above, it should underperform the multi-hypothesis model at the limit.
>
> - _MPSSE and MPSCE also degrade as expected:_ Concerning pose consistency, we computed an MPSSE of 0.4 mm and an MPSCE of 0.8 mm, which are again worse than for the oracle pose. This is of course expected, since Proposition 4.2 in the paper proves that single-hypothesis models (which the aggregation approximates) are bound to lie outside the pose manifold.
>
> [A] Letzelter, V., Perera, D., Rommel, C., Fontaine, M., Essid, S., Richard, G., & Perez, P. Winner-takes-all learners are geometry-aware conditional density estimators. In Forty-first International Conference on Machine Learning.
>
> [B] Du, Q., Faber, V., & Gunzburger, M. (1999). Centroidal Voronoi tessellations: Applications and algorithms. SIAM review, 41(4), 637-676.
>
> [17] Letzelter, V., Fontaine, M., Chen, M., Pérez, P., Essid, S., & Richard, G. (2023). Resilient Multiple Choice Learning: A learned scoring scheme with application to audio scene analysis. Advances in neural information processing systems, 36.
>
> [31] Rupprecht, Christian, et al. "Learning in an uncertain world: Representing ambiguity through multiple hypotheses." Proceedings of the IEEE international conference on computer vision. 2017.

---

> > ### Comment · Reviewer_5BLY · 2024-08-12
> >
> > Thank the authors for addressing my concerns. The theory they prove in the rebuttal is a valuable addition to the contribution of this paper. I have also read the comments from other reviewers and agree that some additional experiments could make this paper more solid. However, I shall vote for acceptance because of the theoretical contributions. If the proofs are guaranteed correct (I only checked the proof sketches, due to limited time and expertise), then the conclusions can be very valuable for the community. Thus, I will keep my rating.

---

### Official Review · Reviewer_B5U1 · 2024-07-12

**Soundness:** 3
**Presentation:** 3
**Contribution:** 3
**Rating:** 7
**Confidence:** 3

**Summary:**

This paper presents a new method to estimate 3D human pose from 2D observations (lifting). To ensure the body symmetry and temporal consistency, the authors disentangle human skeleton to two parts: temporally consistency bone scales and temporally variable bone rotations. The authors use fancy formulas to prove that, minimizing MSE loss could not gurantee manifold consistency. The quantitative and qualitative results on Human3.6m and MPI-INF-3DHP datasets show the superiority of the proposed method.

**Strengths:**

1. The evalution results in this paper is quite impressive, especially the newly proposed consistency metric. Figure 1 clearly shows the superiority of the proposed method.

2. The authors try to prove the theoretical optimal of the proposed method, which is worth encouraging.

**Weaknesses:**

I am not an expert in manifold theory, therefore my questions only relate to human pose estimation.

1. How to constrain the rotation space during training?

2. The pose lifting method is quite similar to Anatomy3D (bone length + rotations). Can I view this paper an multi-hypothesis extension of Anatomy3D? Why?

3. Previous paper "POSE-NDF: MODELING HUMAN POSE MANIFOLDS WITH NEURAL DISTANCE FIELDS" is similar to this paper in concepts. SMPL naturally guarantees bone length symmetry, and the learnable parameters (rotations and shape parameters) are similar to this paper in its functionality. It would be better to cite it.

4. Suppose that, there is a virtual dataset, all 2D human joints are rendered (projected) from strictly symmetric 3D joints, then, could learning the lifting function on this virtual dataset using MSE loss guarantee the results all lie on manifold?

5. (An optional question) The ground truth 3D joints of Human3.6M datasets come from the marker tracking on body surface, which naturally could not guarantee skeleton length consistency. Why learning symmetric bones yields better results (both Anatomy3D and the proposed methods)?

**Questions:**

1. The citation style is weird. They are not NeurIPS style, please correct them.

**Limitations:**

The authors addressed limitations.

---

> ### Author Rebuttal · Authors · 2024-08-07
>
> We thank the reviewer for their comments and provide hereafter our response to their concern, in the same order.
>
> 1. **How to constrain the rotation space during training?**
>
> - _Our method can be adapted to incorporate angle constraints._ This is possible for example if one chooses to use rotation representations where angles appear explicitly (Axis-angle, Euler angles, …). This is not straightforward when using the 6D representations that we chose in this work though, where angles are implicit. We chose these representations because there is a bijection between them and rotations matrices, which presents optimization advantages and allows us to avoid training instabilities (cf. [43] On the continuity of rotation representations in neural networks. CVPR 2019.)
> - _A possible solution:_ If we chose to predict directions instead of rotations (cf. answer to reviewer fpqy Q2), then angles could be easily made explicit in our representation (using spherical coordinates for instance) and we could constrain them to stay within a certain interval by using simple sigmoid activations at the end of our rotations network. This idea could be a nice future extension of our work.
>
> 2. **Is ManiPose a multi-hypothesis extension of Anatomy3D?**
>
> - _Yes, in a sense:_ It is true that, similar to Anatomy3D, ManiPose disentangles limbs length and orientation in order to constrain predicted poses to lie in an estimated manifold. In this sense, we can indeed see it as an extension to the multi-hypothesis setting.
>
> - _But with new theoretical results and a very different message:_ Please note however that, unlike Anatomy3D, we provide theoretical proofs and empirical evidence that both constraints and multiple hypotheses are needed if one wants to optimize joint position error and pose consistency together. This is quite a different message than the one in Anatomy3D paper.
>
> - _And a different way of constraining poses:_ Also note that we use a different representation of limbs orientations than Anatomy3D to constrain our predicted poses (cf. answer to reviewer fpqy Q2). New ablation results in Table 1 of the pdf attached to our main response show that our representation yields better results in the multi-hypothesis setting.
>
> 3. **Concern with SMPL-methods and Pose-NDF citation:**
>
> - _We will eagerly cite Pose-NDF._ We knew and hold this work in high regard. We agree with the reviewer that it is related to our work, as it implicitly learns a plausible sub-manifold of $SO(3)^J$ for poses.
> - _Please note however that Pose-NDF is part of the “body pose and shape estimation” field, which is related but different from 3D human pose lifting._ The objective of most SMPL-based methods is to predict whole 3D body meshes from images instead of 3D joint positions based on 2D keypoints. The task is more challenging, which means that algorithms are heavier and more reliant on optimization-based post-processing. For instance, Pose-NDF estimates 3D meshes in a single image by initializing pose angles and using gradient descent to project them into the learned pose manifold, which is different from doing a simple forward pass with ManiPose.
>
> 4. **Question regarding synthetic data and MSE loss:**
>
> - We can guarantee that if multiple hypotheses _and_ constraints are not used together, then the predictions of the lifting function learned using just MSE loss on the synthetic data proposed by the reviewer _will not_ lie on the manifold (unless the distribution of 3D poses conditioned on 2D poses are Dirac measures, i.e., if there is always a single possible 3D pose projecting into 2D space).
> - This is proved in Proposition 4.2 and shown (with simplified 2-joint and 3-joint articulated objects) in experiments of sections 4.2 and C.2.
>
> 5. **Optional question regarding MoCap:**
>
> - The Human3.6M dataset, as many other datasets, does not contain raw MoCap measurements, but rather post-processed estimations of 3D joint positions obtained through heavy optimization. One of the constraints that is enforced during such post-processing steps is precisely that limb lengths do not vary over time for a given subject. We verify this for ground-truth poses of Human3.6M dataset in figure 5 of our paper (cf. legend).
>
> 6. **Citation style**
> - We apologize for the citation style and will correct it in our revision to comply with the NeurIPS format.

---

> > ### Comment · Reviewer_B5U1 · 2024-08-10
> >
> > The authors rebuttal has clarified my concerns carefully. I also notice that the authors add some experiments according to the comments of other reviewers, which makes the evaluation stronger. If the area chairs could guarantee the correctness of mathematical derivation, I think it would be a good choice to accept this paper.

---

### Official Review · Reviewer_tj82 · 2024-07-13

**Soundness:** 3
**Presentation:** 3
**Contribution:** 3
**Rating:** 5
**Confidence:** 4

**Summary:**

This paper propose ManiPose, a manifold-constrained multi-hypothesis model for 3D human pose lifting. The authors provide empirical and experimental evidence to show that joint position regression leads to inconsistent skeleton lengths. And they propose to predict globally consistent pose scale and individual joint rotations per frame (rather than joint positions) to constrain the predictions to the pose manifold. Empirical results demonstrates that the proposed ManiPose framework improves the pose consistency.

**Strengths:**

* The paper provides valuable theoretical analysis to support their arguments and provides intuitive toy examples to illustrate the ambiguity in pose lifting.
* The paper conducts extensive experiments on H36M and MPI-INF-3DHP datasets.

**Weaknesses:**

* The paper uses a multi-head design to predict multiple hypotheses. This design loses the flexibility of sampling different numbers of hypotheses and limits the maximum number of hypotheses to a small number. This often results in limited hypothesis diversity. In the experimental section, the authors do not provide numerical of visual measurements of hypothesis diversity.
* According to the comparison in Table 4, the manifold constraint proposed in this paper sacrifices MPJPE to improve pose consistency, serving as a trade-off approach between accuracy and consistency. Although the consistency is improved, it lags behind the traditional position regression or manifold regularization in accuracy, and does not bring essential improvement (improve both in accuracy and consistency) compared with these two methods.
* Missing comparison with two recent multi-hypothesis methods. [1] GFPose: Learning 3D Human Pose Prior with Gradient Fields. [2] DiffPose: Toward More Reliable 3D Pose Estimation.

**Questions:**

Please review the Weaknesses Section. If the author can address or respond to the above issues well in the rebuttal stage, I will consider increasing my score.

**Limitations:**

As the authors discussed in the Limitations Section, they used forward kinematics to obtain joint positions, which can lead to error accumulation.

---

> ### Author Rebuttal · Authors · 2024-08-07
>
> We thank the reviewer for their comments and answer here their concerns in the same order.
> 1. **Diversity and fixed number of hypotheses concern:**
>
> - _SOTA oracle MPJPE is evidence of good diversity:_ It is true that ManiPose produces a fixed number of poses per forward pass. While methods based on generative models (GFPose, DiffPose, etc.) typically control the number of samples drawn at test-time, we argue that in predictive tasks, selecting the right number of samples is important to capture the diversity of the plausible targets, i.e., the modes of the conditional distribution. It was indeed proven in prior works that the winner-takes-all (WTA) training scheme, on which ManiPose is based, can achieve an optimal quantization of the target distribution [A] given sufficient data. This diversity in the predicted poses can be measured through Oracle MPJPE performance, for which ManiPose achieves state-of-the-art results.
>
> - _Few hypotheses with scores are more informative:_ Moreover, note that generative methods assign equal weight/likelihood to all sampled poses for a given input. Since they require to sample a large number of poses (~200) to achieve competitive results, it becomes difficult to practically use such an uniform and high-cardinality output in a real scenario. ManiPose, on the other hand, provides more information to the user by predicting just a few relevant consistent poses with their corresponding scores/likelihoods, which is easier to process.
>
> - _Similar coverage of 3D poses distribution as DiffPose:_ In an attempt to quantify the diversity of poses predicted by ManiPose by other means than oracle MPJPE, we have computed the coverage (c.f. [B]) of generated poses over the ground-truth test distribution. For computation cost reasons (it grows quadratically with sample size), we limited our analysis to 5 actions from subject S11 of Human 3.6M. We compare ManiPose to _[2] DiffPose: Toward More Reliable 3D Pose Estimation_, using 5 hypotheses for both, and observe similar diversity on average (c.f. Figure 1 in the pdf attached to the main response).
>
> 2. **Concern with Table 4 and trade-off between consistency and accuracy:**
>
> - _With a single hypothesis, one has to choose between minimizing MPJPE or pose consistency:_ We agree with the reviewer reading of Table 4 and would like to highlight that these are precisely the main messages of our work: 1) in a single-hypothesis setting, constraints will necessarily trade MPJPE for consistency (2nd row) and 2) multiple hypotheses (1st row) are needed to conciliate both MPJPE and consistency metrics.
>
> - _Table 4 is hence empirical evidence supporting our proposition 4.2 and corollary B.1.:_ This result is precisely what makes our approach principled and novel, since there are previous works proposing to constrain predicted poses to estimated manifolds (e.g. Anatomy3D) or proposing to use multiple unconstrained hypotheses (e.g. Diffpose, GFPose, …), but never both together. Please let us know if we misunderstood the reviewers’ point.
>
> 3. **Concern with missing comparison with two recent multi-hypothesis methods:**
>
> - _We will add GFPose and DiffPose to our experimental results._ We thank the reviewer for pointing us to these excellent recent works. We have used their official code and checkpoints to generate 3D poses over the test split of Human3.6M and computed their pose consistency metrics.
>
> - _GFPose delivers worse pose consistency and joint position error in a comparable setting:_ Indeed, we measured an MPSCE of 16.5 mm and an MPSSE of 13.1 mm, considerably worse than ManiPose (MPSCE=0.5 and MPSSE=0.3). In terms of MPJPE, the paper reports an impressive 35.6 mm, but this score makes use of 200 sampled hypotheses, which is 2 orders of magnitude higher than us. In a fairer setting, when using a similar number of hypotheses for both methods, GFPose then stands at 45.1 mm of MPJPE with 10 hypotheses, which is considerably worse than ManiPose (39.1 mm with 5 hypotheses).
>
> - _DiffPose delivers worse pose consistency and competitive joint position error when using their checkpoint and code:_ We computed an MPSCE = 6.1 mm and MPSSE = 5.2 mm for them, which is again inferior to ManiPose. In terms of MPJPE, the authors also report an impressive number of 36.9 mm in their paper for their video model. However, using their official code and checkpoint, and setting the number of hypotheses to 5 as mentioned in the paper, we could only obtain 39.3 mm, which is competitive with ManiPose. Note that there are two other issues on their github page (which we can’t link here) mentioning precisely that the code and checkpoint do not allow to reproduce their reported results.
>
> [A] Letzelter, V., Perera, D., Rommel, C., Fontaine, M., Essid, S., Richard, G., & Perez, P. Winner-takes-all learners are geometry-aware conditional density estimators. In Forty-first International Conference on Machine Learning.
>
> [B] Naeem, M. F., Oh, S. J., Uh, Y., Choi, Y., & Yoo, J. (2020, November). Reliable fidelity and diversity metrics for generative models. In International Conference on Machine Learning (pp. 7176-7185). PMLR.

---

> > ### Comment · Reviewer_tj82 · 2024-08-10
> >
> > Thanks for the authors' response. The response addresses some of my concerns. However, I disagree with the authors' statement that "The diversity in the predicted poses can be measured through Oracle MPJPE performance." MPJPE and diversity reflect two different aspects of the generated results: the accuracy of the best-matching pose and the ability to generate other plausible poses. If we only focus on MPJPE, it aligns more closely with the goal of single-hypothesis pose estimation rather than multi-hypothesis pose estimation.

---

> > > ### Comment · Reviewer_B5U1 · 2024-08-10
> > >
> > > I think the authors' claim in the rebuttal rooted in the conclusion of "Winner-takes-all learners are geometry-aware conditional density estimators". Maybe the authors could elaborate on this point by showing how to map the conclusion of "winner-takes-all ..." paper with the claim here. From my point of view, the diversity has been demonstrated in Figure.6. Such visualization of diversity has also been used in previous papers like HMFormer.
> > >
> > > In fact, I think this claim is not totally unreasonable. For example, for a side view person whose left arm is totally occluded, a good predicted distribution (with proper or enough diversity) should cover the pose as close to GT as possible, which would result in a better MPJPE in winner-takes-all evaluation scheme. If it is not overfitting, then the predicted distribution must have a good diversity, at least covering the one close to GT. If the training dataset is large enough, it can not overfit for the occluded parts. Maybe the authors could demonstrate such cases by visualization, which would be much more convincing.
> > >
> > > I would also like to listen to the advice of other reviewers and the authors about this good question.

---

> > > ### Author Response · Authors · 2024-08-10
> > > **Answer to comments of reviewers tj82 and B5U1**
> > >
> > > We thank the reviewers for their prompt answers. Reviewer B5U1 is correct in interpreting Figure 6 and in saying that our answer is connected to [A] "Winner-takes-all learners are geometry-aware conditional density estimators". But we understand reviewer tj82's concern. The reason why we say that **oracle MPJPE** is one way of assessing diversity is because it writes as
> > > $$\frac{1}{N} \sum_{i} \min_k \ell(f_k(x_i), \mathrm{p}\_i),$$
> > > which **is an approximation when $N$ is large of the quantization error**, also known as distortion:
> > > $$\int_{\mathcal{X} \times \mathcal{P}} \min_k \ell(f_k(x), \mathrm{p}) \rho(x, \mathrm{p}) \mathrm{d} x \mathrm{d} \mathrm{p},$$
> > > where $\ell$ is the average joint-wise $L2$ distance in our case. The latter is traditionally used to measure the efficiency of an estimator in summarizing a distribution with few representatives, commonly used to study the K-means estimator for example [C].
> > >
> > > So the fact that we achieve better oracle MPJPE than methods requiring a large number of hypotheses (GFPose, D3DP, Wehrbein et al., …) shows that **ManiPose has better quantization properties** than the latter, i.e., it is more efficient in summarizing the diversity of the conditional distribution with fewer representatives/hypotheses.
> > >
> > > More practically:
> > > 1. the extreme case of a model with no diversity at all (e.g., predicting $K$ times the same pose) would lead to an oracle MPJPE = vanilla MPJPE of its single-hypothesis version. This is not what we obtain in our ablation study of Table 4 (1st vs last rows).
> > > 2. In the opposite extreme case, a naive way to obtain $K$ very diverse hypotheses would be to use a regular grid of the pose space $\mathcal{P}=\mathbb{R}^{3 \times J}$. The latter learns nothing and is uninformative, but could still achieve better oracle MPJPE if given an unrealistically large number of hypotheses (cf [A] equation 16).
> > > 3. As shown in [A], the winner-takes-all learning scheme, used in ManiPose, allows it to sit between these extreme cases by learning an adaptive “grid” made of a few hypotheses, capturing the geometry of the underlying conditional distribution. Results on its quantization optimality can be found in section 5.2 of [A] for example.
> > >
> > > Of course, oracle MPJPE is not the only way of assessing diversity, which is why **we have provided in our rebuttal additional results** measuring the coverage of ManiPose, which corresponds to the ratio of ground-truth poses whose neighborhood contains at least one generated pose. It is a common metric used in the literature on generative models to analyze diversity.
> > >
> > > [C] Pages, Gilles, and Jacques Printems. "Optimal quadratic quantization for numerics: the Gaussian case." (2003).

---

> > > > ### Comment · Reviewer_tj82 · 2024-08-11
> > > >
> > > > Thanks for the authors' response and Reviewer B5U1's comments. I acknowledge that ManiPose demonstrates better quantization properties compared to generative model-based methods like GFPose and D3DP. I suggest that the authors include this comparison and emphasize this point in their revision, as it highlights a unique advantage of their approach.
> > > >
> > > > However, I believe that better summarization does not necessarily imply better diversity. I recommend that the authors avoid making such disputed claims in their paper. I also welcome input from other reviewers on this matter.

---

> > > > > ### Author Response · Authors · 2024-08-11
> > > > >
> > > > > We thank the reviewer for taking into account our explanations. We agree that the definition of “diversity” used is indeed important to specify, and will clarify our quantization results in the revised manuscript.

---

> > > > > > ### Comment · Reviewer_tj82 · 2024-08-13
> > > > > >
> > > > > > Thanks for the authors' response. While I still have concern about the diversity of the results, I recognize that the proposed method offers better summarization ability compared to GFPose and D3DP. Therefore, I am inclined to raise my score. I recommend that the authors include additional metrics related to diversity in their paper and avoid using "Oracle MPJPE" as a proxy for measuring diversity.

---

### Author Rebuttal · Authors · 2024-08-07

We thank the reviewers for their work. We provide answers to all their concerns individually, referring sometimes to the pdf attached to this general answer.

We would like to highlight that our rebuttal includes:
- a new theoretical result, together with its proof sketch,
- a new ablation study related to our rotations representation,
- the evaluation of two new baseline methods,
- and new evaluations of our method in terms of diversity and consistency of aggregated poses.

---

### Decision · Program_Chairs · 2024-09-25

**Decision:**

Accept (poster)

**Comment:**

This paper presents a manifold-constrained multi-hypothesis model for 3D human pose lifting.  To ensure the body symmetry and temporal consistency, the method disentangles human skeleton to two parts: temporally consistency bone scales and temporally variable bone rotations. The method then predicts skeletal parameters so that the predicted 3D poses in a sequence are constrained to one smooth manifold. The paper provides theoretical arguments regarding the benefits of enforcing such structural priors in the prediction. The proposed method is validated on Human3.6M and MPI-INF-3DHP datasets. Although theoretical analysis to support the arguments in the paper is appreciated, the reviewers raised concerns regarding diversity and fixed number of hypotheses, unclear advantage of multi-hypothesis methods, undetailed explanation about the method, trade-off between accuracy and consistency, missing comparison, and difference from Anatomy3D and Pose-NDF.  The authors addressed in their rebuttal most of the concerns by providing new arguments, new experiments and new theoretical analysis. During post-rebuttal discussion, the authors further clarified the contributions of the paper, and convinced three of the four reviewers.  The reviewer tj82 suggested to specify the definition of diversity, which AC thinks should be fixed.  The review fpqy, on the other hand, was not satisfied the feedbacks from the authors.  He/She keeps concern on the positioning of the paper, meaning that theoretical contribution or 3D pose lifting application is unclear.  It was also pointed out that the new theoretical results provided in the rebuttal are obvious.  AC does not agree with the reviewer fpqy in the sense that even obvious theoretical results are sometimes valuable.  This is because results in good harmony with our intuition bring to us the strong ground of the method rather than heuristically/empirically developed methods. AC also thinks the balance between theoretical analysis and the proposed lifting method is good and does not reduce the strengths of the paper. This paper should be accepted, accordingly.